# TOWARDS UNIVERSALITY: STUDYING MECHANISTIC SIMILARITY ACROSS LANGUAGE MODEL ARCHITECTURES

**Junxuan Wang**[1]    **Xuyang Ge**[1]    **Wentao Shu**[1]    **Qiong Tang**[1]    **Yunhua Zhou**[1]
**Zhengfu He**[1,2]    **Xipeng Qiu**[1,2,*]

[1]OpenMOSS Team, School of Computer Science, Fudan University
[2]Shanghai Innovation Institute
`junxuanwang21@m.fudan.edu.cn`
`{zfhe19,xpqiu}@fudan.edu.cn`

## ABSTRACT

The hypothesis of *Universality* in interpretability suggests that different neural networks may converge to implement similar algorithms on similar tasks. In this work, we investigate two mainstream architectures for language modeling, namely Transformers and Mambas, to explore the extent of their mechanistic similarity. We propose to use Sparse Autoencoders (SAEs) to isolate interpretable features from these models and show that most features are similar in these two models. We also validate the correlation between feature similarity and *Universality*. We then delve into the circuit-level analysis of Mamba models and find that the induction circuits in Mamba are structurally analogous to those in Transformers. We also identify a nuanced difference we call *Off-by-One motif*: The information of one token is written into the SSM state in its next position. Whilst interaction between tokens in Transformers does not exhibit such trend.

## 1 INTRODUCTION

The field of mechanistic interpretability seeks to reverse engineer real-world neural networks into interpretable *features* and *circuits*. Due to the massive societal importance of Transformer language models (Vaswani et al., 2017; Brown et al., 2020; Ouyang et al., 2022; Sun et al., 2024), existing literature in this field mostly focuses on Transformers (Elhage et al., 2021; Olsson et al., 2022; Wang et al., 2023). This poses a natural research question:

**Will mechanistic findings in one model transfer to others?** The answer to this question may tell interpretability researchers how much of their current understanding will be of use in the future. The conjectural transferability of mechanistic understanding of a specific model is called the *Universality Hypothesis* (Olah et al., 2020; Chughtai et al., 2023). Concretely, there are several aspects of *Universality* that may be of interest: (1) **Training:** Will models with different training settings converge to similar representational structures? (2) **Model Size:** Are features and circuits in smaller models present in larger models? If so, what do larger models build further on these structures? (3) **Model Architecture:** How similar is the mechanism in models of different architectures? Does the universal part teach us more general motif of neural networks and language modeling? Does the divergence indicate the pros and cons of each architecture? (4) **Multilingualism / Cross-Modality:** Are neural networks able to obtain similar knowledge from different languages or modalities?

This work mainly focuses on Architectural *Universality*. We believe the answer to this, if it exists, lies somewhere in the middle of the spectrum. It may not be the case that all models are isomorphic, nor that they implement totally distinct algorithms. If the commonality of *features* and *circuits* across models of different architectures is revealed, we may be more confident to generalize some of our current mechanistic findings in pretrained Transformer language models to perception of the

---

*Corresponding author.

task of language modeling. Pragmatically, such analysis also sheds light on whether one family of models fail to represent some certain type of features or implement some circuits present in another. This may help provide a microscopic comparison among model architectures (Vaswani et al., 2017; Peng et al., 2023; Sun et al., 2023; Gu & Dao, 2023).

To this end, we mainly investigate two of the most popular types of models for language modeling, namely (decoder-only) Transformers and Mambas (Gu & Dao, 2023), a family of models based on State Space Models (SSMs).

We manage to bridge these two models with features extracted by Sparse Autoencoders (SAEs) in an unsupervised manner. Qualitative and quantitative experiments show that these two family models learn a lot of similar features. We also utilize our SAEs to investigate induction circuits (Olsson et al., 2022) in Mamba, which are also highly analogous to ones in Transformers.

However, the mechanism in these two models is nuanced by what we call *Off-by-One Preference* in Mamba. We find that Mamba often writes the information of a specific token into the SSM state in its next token. We show that this is done by the local convolution layer.

Our main contribution can be summarized as follows. **(1)** To the best of our knowledge, this is the first work to look for linear, interpretable features with Sparse Autoencodersand quantitatively evaluate the similarity between Transformer and Mamba features (Section 4). **(2)** We propose a novel metric to isolate and quantify feature universality with respect to architectural changes (Section 4 and 5). **(3)** We perform circuit analysis in Mamba and qualitatively reveal its structural analogy and some nuances compared with Transformer circuits (Section 6).

## 2 RELATED WORK

**Representational Universality.** The concept of *Universality* in mechanistic interpretability was introduced by Olah et al. (2020) and has been extensively studied as representational universality (Klabunde et al., 2023; Paulo et al., 2024). Raghu et al. (2017) and Kornblith et al. (2019) measured representation similarity using canonical correlation analysis, while other methods (Ding et al., 2021; Khrulkov & Oseledets, 2018; Boix-Adserà et al., 2022) quantify similarity across neural networks. Efforts also include examining neuron correlations in identical models with different seeds (Li et al., 2016; Wang et al., 2018; Gurnee et al., 2024), with recent work suggesting convergence across modalities (Huh et al., 2024).

**Circuit Universality.** Circuit-based mechanistic interpretability was pioneered by Cammarata et al. (2020); Olah et al. (2020); Elhage et al. (2021), investigating whether features and circuits are analogous across models. Olah et al. (2020) showed early layer similarity in vision models, with further studies extending to MLPs and Transformers on algorithmic tasks (Chughtai et al., 2023).

**Sparse Autoencoders.** Neuron-based analysis is often hampered by polysemantic neurons (Elhage et al., 2023; 2022b). SAEs (Bricken et al., 2023; Cunningham et al., 2023; Templeton et al., 2024) extract monosemantic sparse codes from activations, providing interpretable features. SAEs have been applied to language models (Rajamanoharan et al., 2024; Ge et al., 2024) and tasks like Othello move prediction (He et al., 2024).

**Mamba Interpretability.** Mamba (Gu & Dao, 2023; Dao & Gu, 2024), a family of models based on State Space Models (SSMs), is relatively new in interpretability studies. Ali et al. (2024) found that Mamba implicitly attends to previous tokens via its B, C matrices. Studies have explored activation patching on Mamba (Meng et al., 2022; Sharma et al., 2024), with a unified view of Transformers and Mambas emerging (Zimerman et al., 2024; Dao & Gu, 2024).

## 3 PRELIMINARIES

### 3.1 MODELS

We choose to study Pythia-160M (Biderman et al., 2023) and an open-source version of Mamba-130M[1]. Pythia-160M uses an untied word embedding and unembedding, leading to about 30 million (i.e. vocabulary size by hidden dimension) more parameters. They adopt the same tokenizer and both are trained on the Pile dataset (Gao et al., 2021). Unless otherwise specified throughout this paper, we refer to these two models as Pythia and Mamba. Since Mamba adopts a GAU-type architecture (Hua et al., 2022) which implements both token and channel mixing in one Mamba block, it has twice the depth (i.e. 24 layers) as Pythia (i.e. 12 layers). We also include experimental results for the 2.8B models of both families in Appendix D.5 to demonstrate the generalizability of our findings.

### 3.2 A BRIEF MATHEMATICAL FRAMEWORK OF MAMBA CIRCUITS

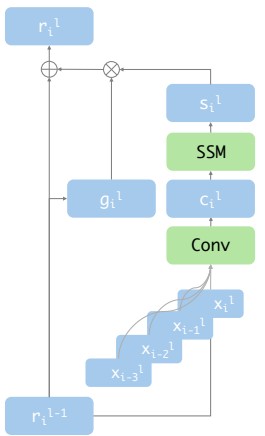

Figure 1: A simplified Mamba block.

As shown in Figure 1, Mamba architecture incorporates a number of Mamba Blocks. This architecture adopts a layer-stacking approach, akin to the stacking of Transformer blocks in Transformer models with residual connections employed between layers. Each block takes in a copy of the residual stream $r_i^{(l-1)}$, which is used to independently compute the gate output $g_i^{(l)}$ and SSM output $s_i^{(l)}$, which are merged with a Hadamard product and written into the residual stream with a linear transformation $W_o^{(l)}$:

$$r_i^{(l)} = r_i^{(l-1)} + W_o^{(l)}(s_i^{(l)} \cdot g_i^{(l)}). \tag{1}$$

The gate output $g_i^{(l)}$ is simply a linear projection followed by a non-linear activation:

$$g_i^{(l)} = \text{SiLU}(W_g^{(l)} r_i^{(l-1)}). \tag{2}$$

State space model outputs stand as the core of Mamba computation, which performs inter-token information mixing. Specifically, a local convolution layer with a window size of $d_{\text{conv}}$ is applied to a linear projection of $r_i^{(l-1)}$:

$$x_i^{(l)} = W_{in}^{(l)} r_i^{(l-1)}; \quad (3) \qquad c_i^{(l)} = \text{SiLU}(\text{Conv1D}(x_i^{(l)}, x_{i-1}^{(l)}, ..., x_{i-d_{\text{conv}}+1}^{(l)})). \tag{4}$$

After this local information aggregation, a subsequent SSM layer performs global token mixing, which combines its current state $h_{i-1}^{(l)}$ and updates to $h_i^{(l)}$ with $c_i^{(l)}$. Since all variables and computation are restricted at layer $l$, we omit the superscripts in the following equations.

$$h_i = F_a(c_i) \cdot h_{i-1} + F_b(c_i), \quad (5) \qquad s_i = h_i(W_c c_i) + W_d \cdot c_i. \tag{6}$$

Concretely, $F_a$ and $F_b$ implement the data-dependent Selection Mechanism for Mamba, which are to some extent similar to *forget gate* and *input gate* in LSTMs, respectively. $W_c$ and $W_d$ decide the output of this SSM layer, much like an *output gate*. Such analogy provides analytic simplicity but also vagues a number of details of Mamba architecture. We refer readers to Gu & Dao (2023) and Dao & Gu (2024) for further details.

---

[1]https://huggingface.co/state-spaces/mamba-130m

## 3.3 SPARSE AUTOENCODERS

Our SAEs have only one hidden dimension larger than the input dimension (i.e. $F > D$), with the training objective of reconstructing any given model activation and an L1 penalty on its hidden layer to incentivize sparsity. An SAE can be formulated as follows:

$$f_i(\mathbf{x}) = \text{ReLU}(\mathbf{W}^{enc}_{i,:}\mathbf{x} + b^{enc}_i), \quad (7) \qquad \hat{\mathbf{x}} = \sum_{i=1}^{F} f_i(\mathbf{x})\mathbf{W}^{dec}_{:,i}, \qquad (8)$$

where $\mathbf{x} \in \mathbb{R}^D$ is the hidden activation decomposed with $F$ features. $\mathbf{W}^{enc} \in \mathbb{R}^{F \times D}$ and $\mathbf{W}^{dec} \in \mathbb{R}^{D \times F}$ are the encoder and decoder of an SAE. $\mathbf{x}$ is linearly mapped onto the feature space by the encoder and a bias $b^{enc}$, followed by a ReLU to ensure that $f(\mathbf{x})$ (i.e., feature activations) is nonnegative.

SAEs are trained to reconstruct the original activation with a linear combination of the decoder columns (i.e. features), determined by $f(\mathbf{x})$. We also set an L1 sparsity constraint on $f(\mathbf{x})$ to obtain a sparse coding of each activation. Our training technique is known as vanilla SAEs. We include results of more performant variants, i.e., TopK-SAEs (Gao et al., 2024).

# 4 CROSS-ARCHITECTURE UNIVERSALITY WITH FEATURE SIMILARITY

## 4.1 IN SEARCH OF INTERPRETABLE PRIMITIVES

We seek to address this *Universality* problem in a **fine-grained** manner by comparing models with an independent decomposition of the activation space. We see this kind of microscopic comparison first pursued by analyzing correlation between individual MLP neurons in models with the identical architecture (Li et al., 2016; Gurnee et al., 2024).

However, there is hardly a counterpart of MLP neurons in Mamba. This suggests the necessity to utilize tools invariant to architectural changes to study architecture-agnostic similarity. To this end, we leverage a trending tool in interpretability called Sparse Autoencoders to extract more monosemantic units (i.e., features) from the models' activation.

## 4.2 MATCHING FEATURE PAIRS ACROSS ARCHITECTURES

### 4.2.1 TRAINING SPARSE AUTOENCODERS

We trained SAEs in the residual stream after each Transformer block in both models, getting 12 and 24 SAEs in Pythia and Mamba, respectively (See Section 3.1). Both models have a hidden dimension $D = 768$. We set the hidden dimension of SAEs to $F = 32D = 24576$ for all Sparse Autoencoders (SAEs). We refer readers to Appendix A for more details of our SAEs.

### 4.2.2 MAIN EXPERIMENTS

SAE features are analogous to MLP neurons in that they both linearly correspond to a direction and has (almost) nonnegative activations. Thus, we can follow Li et al. (2016); Bau et al. (2019) and Gurnee et al. (2024) to evaluate feature similarity by identifying feature pairs activated on the same inputs.

We iterate over 1 million tokens sampled from SlimPajama and save the activation of each SAE feature in both models. For each SAE feature in Pythia, we computed the maximum Pearson correlation among all features for all SAEs across all layers in Mamba, which we refer to as the Max Pairwise Pearson Correlation (MPPC).

$$\rho^{p \to m}_i = \max_j \frac{\mathbb{E}[(\mathbf{v}^p_i - \mu^p_i)(\mathbf{v}^m_j - \mu^m_j)]}{\sigma^p_i \sigma^m_j}, \qquad (9)$$

where $\mathbf{v}^p_i, \mathbf{v}^m_j \in \mathbb{R}^N$ are the activation pattern of the $i$-th feature in Pythia and the $j$-th feature in Mamba. We compute the mean and variance for each pattern, denoted by $\mu^p_i, \mu^m_j, \sigma^p_i$ and $\sigma^m_j$.

It is worth noting that the superscript in $\rho_i^{p \to m}$ implies that we are looking for the Mamba feature most correlated to the $i$-th Pythia SAE feature. However, this operator is not commutative. We use this metric throughout this paper as our main experiment. To show that this unilateral analysis does not affect our conclusion, we include all reverse results, i.e., $\rho^{m \to p}$ in Appendix C.

### 4.3 BASELINES AND SKYLINES

#### 4.3.1 RANDOM AND NEURON BASELINE

When computing MPPC, its value reflects the maximum from numerous samples, introducing a positive bias. Additionally, SAE properties like sparsity constraints can further shift the distribution towards a higher value. To mitigate this, we established a baseline by calculating MPPC for each Pythia feature against a random SAE with matching feature count and sparsity level (by masking all but the Top K activating features) to the Mamba SAE.

As noted in Section 4.1, it does not make intuitive sense to compare architectures in *neuron basis*. We empirically validate this by computing the correlation introduced in Equation 9 between all Pythia MLP neurons and all Mamba post-SiLU activations.

#### 4.3.2 HIERARCHY OF DIMENSIONS OF VARIATION

Studying universality across neural network architectures with SAEs implicitly discusses a number of dimensions of variation. We believe there are three levels of variables with some relation of inclusion.

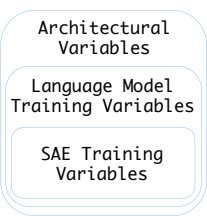

Figure 2: Hierarchy of invariance.

- **SAE Variables:** (Innermost area in Figure 2) It is important to ensure that SAEs actually learn its latent features in a *replicable* manner. That is, we have to sweep hyperparameters for training SAEs on the same activation space and make sure they mostly learn similar features. SAE features can serve our purpose only if SAE itself do not introduce much randomness in its learned features. Such invariance property needs to be validated with changes to *SAE initialization, training batch size, learning rate / schedules, number of training tokens and order of training data, etc.*

- **Language Model Variables:** (Middle area) Similarly, language model training is a stochastic progress and may further introduce uncertainty to the model's internal. Model variables here include *data distribution, training tokens elapsed and training hyperparameters similar to SAE variables*. Training SAEs on different models must give different results since at least their training data (i.e. model activations) are different.

- **Architectural Variables:** (Outermost area) Most aforementioned model variables, especially ones related to initialization and training, are implicitly included in architectural changes. For example, it is certain that we cannot find an identical initialization or training trajectory for Mamba models and Pythia models.

To this end, we propose two skylines to indirectly estimate the marginal effect of each area by ablation.

**Skyline 1: Different Models of Identical Architecture (Model Seed Variant).** For each feature in Pythia, we look for similar features in another Pythia model trained with a different random seed. This changes model initialization and training data order. The intuitive motivation of this is that a Mamba model has all differences the seed variant has plus the architectural change.

**Skyline 2: SAEs Trained on the Same Language Model (SAE Seed Variant).** We also trained another set of SAEs on the same Pythia model as a higher skyline. This setting further constrains the language model initialization and training factors. The only difference lies in SAE training. The discrepancy between SAEs trained on the same model tells us how convergent SAE features are.

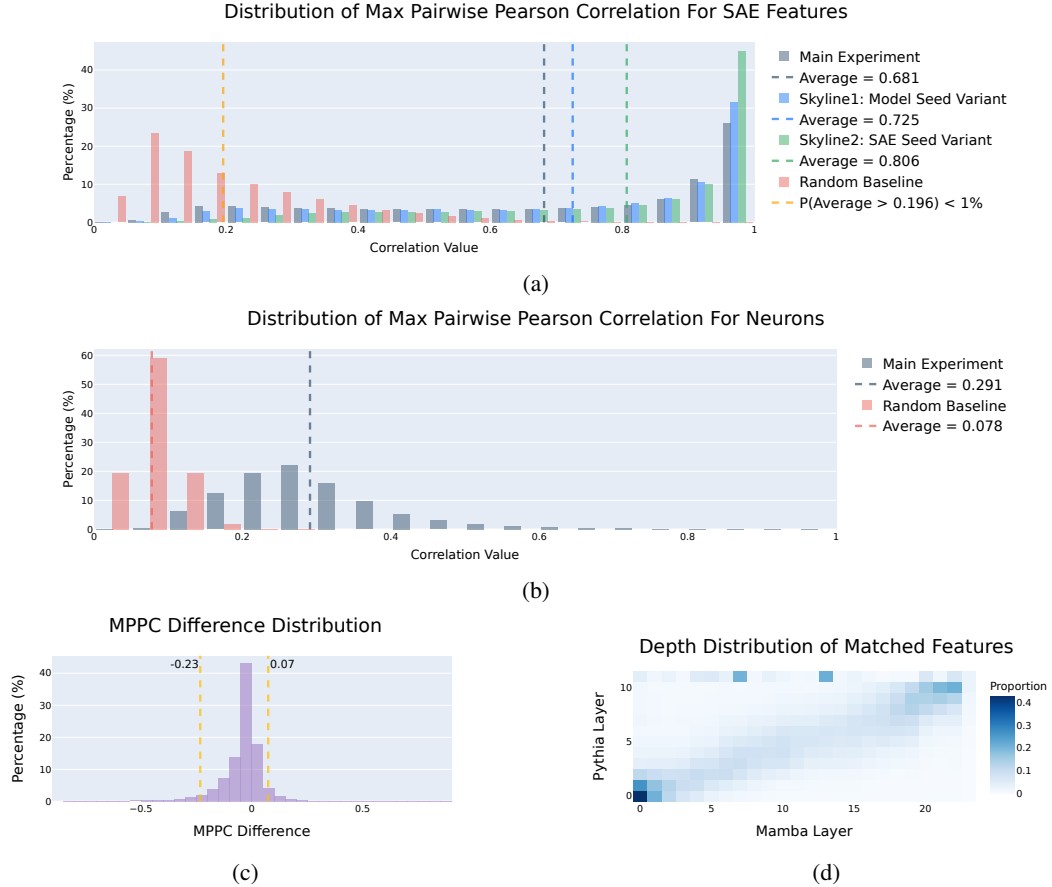

(a)

(b)

(c)                                                      (d)

Figure 3: (a) MPPC distribution for SAE features: main experiment, model seed variant skyline, SAE seed variant skyline, and random baseline. (b) MPPC distribution for neurons: main experiment vs. random baseline. (c) Distribution of MPPC differences for SAE features, computed as main experiment minus model seed variant skyline. (d) Layer-wise frequency of best-matching feature pairs, mostly concentrated near the diagonal (similar depth). The anomaly in Pythia's final layer may result from significant residual stream differences, reducing SAE quality.

## 4.4   RESULTS

Figure 3a presents the results of all MPPC experiments on SAE features, namely random baseline, Pythia-Mamba SAE similarity (main experiment), model seed skyline and SAE seed skyline. Figure 3b presents the results of random baseline and main experiment on neurons.

**Significant Cross-Architecture Similarity of SAE Features**   Neurons show much lower similarity across architectures, with fewer than 15% achieving MPPC > 0.4, compared to over 55% of SAE features exceeding 0.7. This supports the hypothesis that comparing privileged directions across models is uninformative (Elhage et al., 2023). In contrast, SAE features in the main experiment significantly outperform the random baseline, with the mean MPPC value exceeding the 99th percentile of the mean MPPC distribution[2] for the random baseline, demonstrating statistical significance. These results highlight the strong cross-architecture similarity of SAE features.

**Cross-Architecture Similarity Slightly Falls Short of Skylines.**   Cross-arch SAE MPPC has an average of 0.681, compared to 0.725 for model seed variant and 0.806 for SAE seed variant. More than 25% Pythia features find their matched feature in Mamba with an MPPC greater than 0.95.

---

[2]We repeated the baseline 16 times and calculated the mean MPPC for each repetition to fit this distribution (assuming the mean MPPC follows a normal distribution).

We illustrate the distribution of the difference between our Cross-arch SAE MPPC and the model seed variant skyline in Figure 3c, i.e. $\rho_i^{p \to m} - \rho_i^{p \to p'}$ where $i$ indexes a feature in Pythia, ranging from 0 to $12 * 24576$[3]. One can interpret this as the marginal effect of architectural changes. 90% of Pythia features fall in the interval of $-0.23$ to $0.07$ under this metric. More than 60% features have near-zero MPPC difference(absolute value $< 0.05$), which means in our case, they are consistently present after architectural changes.

Our findings provide strong evidence that SAE features exhibit universality across architectures. We further validated our method and conclusions in additional settings, including alternative model architectures, larger models, varied SAE hyperparameters, advanced SAE designs, and diverse datasets (see Appendix D for details). However, using SAEs for feature decomposition is a relatively new approach, and these conclusions rely on specific assumptions. For a detailed discussion of limitations, refer to the limitation section(Appendix 7).

### 4.5 DEPTH SPECIALIZATION

Another interesting phenomenon is that SAE features form similar hierarchy in different architectures. Our findings extend the scope of neuronal depth specialization (Gurnee et al., 2024) where the authors find universal neurons to be present at near layers. As illustrated in Figure 3d, if a feature in Pythia is located in the $l$-th layer out of 12 layers, its matched feature in Mamba is most likely to appear around layer $2l$ in Mamba (out of 24 layers). This provides evidence for a more macroscopic conjecture of language model, i.e. the general structure of a specific language model internals is a *stretched* or *scaled* version of a universal motif (Elhage et al., 2022a; Huh et al., 2024).

## 5 A COMPLEXITY-BASED INTERPRETATION OF SIMILARITY DIFFERENCE

We demonstrate in Section 4 quantitatively that a great number of Pythia features can be paired with one Mamba feature in terms of MPPC. It is further required to map such statistics onto our hypothetical *Universality*. Anecdotally, we obeserve that a higher MPPC feature pair *does not necessarily mean more semantic similarity* than lower ones due to the influence of *feature complexity*. This offers us a new perspective of the aforementioned MPPC difference between our Cross-Arch SAE MPPC and the seed variant skyline (Section 4.3): such difference can be deemed as cancelling out the effect of *feature complexity*.

We manually inspected a number of matched feature pairs across architectures and seed variants. An intriguing trend is summarized as follows and also illustrated in Figure 4.

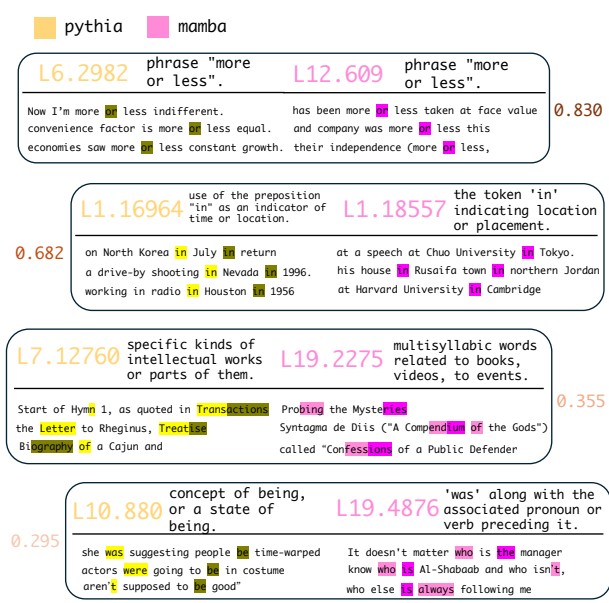

Figure 4: We present four cases of feature pairs with different Pearson correlation values beside. To the right of each feature index is the auto-interpretation result generated by GPT-4o, with activation examples shown below.

---

[3]This is the number of layers multiplied by number of features in each SAE

## 5.1 CASE STUDY

**Higher MPPC pairs are mostly simple features.** Features with a maximum pairwise Pearson correlation close to 1 are typically very simple, activating exclusively on individual tokens. These include a wide range of linguistic units such as all parts of speech, word prefixes, suffixes, letters, punctuation marks, and digits (the first row in Figure 4).

**Lower MPPC pairs tend to be more complex.** As the *complexity* of features increases, the MPPC scores of most matched pairs fall in the interval $0.3 < \rho_i^{p \to m} < 0.8$. Intuitively, if both features fire at the same high-level structure, it is more likely that the activation magnitude of one feature is larger for some specific examples. While another feature may instead prefer some other instances, leading to inconsistency in firing strength but agreement in theme. We provide further discussion in Section 5.2.

**Ultra-Low MPPC pairs are mostly uninterpretable.** Features with extremely low maximum pairwise Pearson correlation are typically uninterpretable and considered meaningless, possibly arising from noise introduced during SAE training or model training. We provide further discussion in Section 5.2.

## 5.2 QUANTIFYING FEATURE COMPLEXITY AND MONOSEMANTICITY

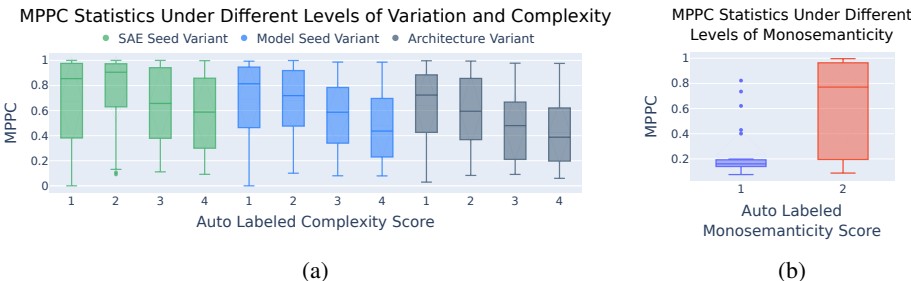

Figure 5: (a) Distribution of MPPC over auto-labeled complexity scores ranging from 1 (simple) to 4 (complex). Both Model Seed Variant and Cross-Arch SAE MPPC exhibit correlation, while one in SAE Seed Variant is weaker since the mean MPPC at score 1 is smaller than that at score 2. (b) Distribution of MPPC over monosemanticity scores ranging from 1 (Polysemantic) to 2 (Monosemantic). Monosemanticity scores almost divide MPPC into two interval categories at about 0.2.

We are aware that complexity and monosemanticity is a somehow abstract property of a feature. To effectively validate our conjecture, we utilize GPT-4o to automatically evaluate the complexity and activation consistency of each feature's activation pattern.

We mainly follow Bills et al. (2023) and Cunningham & Conerly (2024) to prompt GPT-4o to score the degree of complexity and activations consistency of each feature. The activation pattern on the first 10 top activating samples of each feature are fed into GPT-4o in a structured format (Appendix E). It then returns an integer score ranging from 1 to 4. For complexity, a score of 1 indicates the simplest features, while a score of 4 represents the most diverse contexts with a coherent theme. We validated the effectiveness of GPT-4o's scoring by comparing it against human expert evaluations (Appendix I).

We observe clear correlation between automatically labeled complexity scores and seed variant skyline MPPC, i.e. $\rho_i^{p \to p'}$, as well as Cross-Arch SAE MPPC, as shown in Figure 5a. Statistically, the simpler a Pythia feature is, the more likely it matches another feature in model seed variant or Mamba with a high Pearson Correlation score.

This offers another interpretation of isolating architectural variables with MPPC difference proposed in Section 4.3.2. Cross-Arch SAE MPPC reflects a mixture of several variables. One of them is the semantic complexity of features, which is present in both Cross-Arch SAE MPPC and seed variant

skyline MPPC. Subtracting the latter from the former can be deemed as **cancelling out the effect of feature complexity**.

The trend for the Monosemanticity Score (Figure 5b) is even more pronounced, with a clear boundary around 0.2. Features with MPPC values below this threshold are highly likely to be uninterpretable. We refer readers to Appendix F for some cases.

### 5.3 IMPLICATIONS TO MODEL DIFFING METHODS

The complexity issue of MPPC confounds the true different features and those matched but with an abstract theme. Thus, we fail to identify any architecture-specific feature. Our complexity viewpoint of MPPC raises a broad concern for cross-architecture model diffing. For example, a recent promising model diffing method crosscoder (Lindsey et al., 2024) is also prone to false positive of divergent features, where features activating on the same concept might be identified as different ones due to their respective preference for specific instances. This suggests the need for sanity checks of subsequent model diffing methods.

## 6 ZOOMING OUT TO INDUCTION CIRCUIT UNIVERSALITY

We also want to look into a larger scale of universality by studying the circuits in Mamba and Transformers. We choose to start from a relatively simple and well-studied circuit named induction circuit (Olsson et al., 2022).

The in-context learning behavior is when sequences with pattern $[A][B]...[A]$ appear, the model is very likely to output $[B]$ in the subsequent step to complete the 2-gram with knowledge from the past. It has been shown that this is closely related to a kind of an algorithm mainly implemented by two attention heads in Transformers and reoccurs broadly in many Transformer language models (Olsson et al., 2022; Todd et al., 2024).

It has been already demonstrated by Fu et al. (2023); Gu & Dao (2023), *inter alia*, that Mamba is also capable of in-context learning. An important curiosity is how Mamba implements this algorithm since it does not have attention heads. We find that Mamba employs a similar mechanism to Transformers, but with some slight differences in its implementation.

### 6.1 IDENTIFYING INDUCTION CIRCUITS IN MAMBA

We employ a circuit discovery method called path patching (Wang et al., 2023) to search for which components in Mamba directly affecting the output logits, as shown in (Figure 6).

Path patching replaces the activations along specified paths and allows this influence to propagate along the allowed path then observe the difference of interest. This measures the counterfactual influence of the specified paths only through allowed paths. The data results for all experiments in this section are provided in Appendix G.

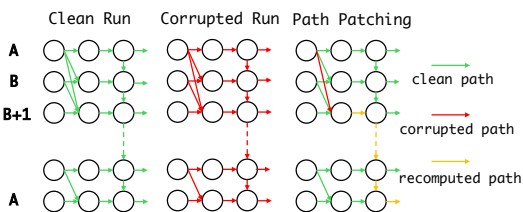

Figure 6: Path patching includes three times of forward pass. We patch a specified path (red edges) with a corrupted path and uses the recomputed activations only through the paths of interest (yellow edges). Activations through other paths (green edges) are fixed to the same as the clean run.

**Layer 17 SSM States Are Key in Induction.** We label key positions in $[A][B]...[A]$ as $A_1$, $B_1$, and $A_2$, and start with a counterfactual input $[A][B']...[A]$. Path patching is applied to $h_{A_2-1}^{(l)}$ for $l$ layers (0-24), allowing the influence to propagate except through other SSM states. Patching at layer 17 results in the largest logit drop, with layer 20 having a weaker effect and other layers mostly irrelevant.

**Key Information Flows into SSM States at Token B+1.** We next patch components of $h_{A_2-1}^{(17)}$, which depends on $c_{A_2-1}^{(17)}, \ldots, c_0^{(17)}$. Results (Figure 7a) show patching $c_{B_1+1}^{(17)}$ significantly reduces the logit for $B$, with no substantial changes at other positions.

**Local Convolution Aggregates A and B into B+1.** Finally, we patch components of $c_{B_1+1}^{(17)}$, a linear combination of $x_{B_1+1}^{(17)}$, $x_{B_1}^{(17)}$, $x_{A_1}^{(17)}$, and $x_{A_1-1}^{(17)}$ (Equation 4). Patching $x_{B_1}^{(17)}$ or $x_{A_1}^{(17)}$ with corrupted inputs, $[A][B']...[A]$ or $[A'][B]...[A]$, significantly affects the final logits, indicating that the short convolution layer aggregates information from both $[A]$ and $[B]$ into the SSM states.

## 6.2 MAMBA AND TRANSFORMER IMPLEMENT THE SAME INDUCTION ALGORITHM

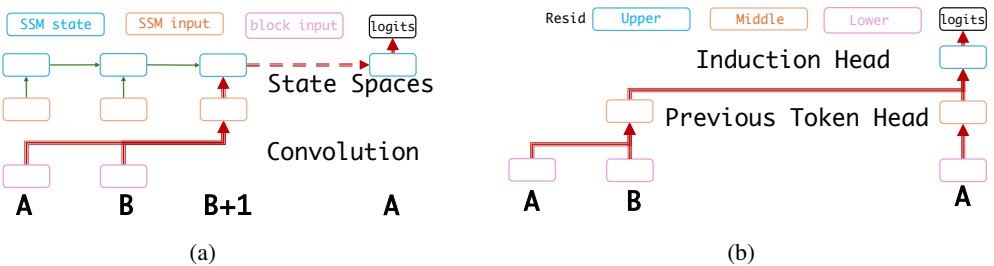

Figure 7: Induction Circuit in (a) Mambas. (b) Transformers.

We have abstracted the induction circuit in the Mamba model (Figure 7a), while in Transformers, this is implemented through a previous token head and an induction head (Figure 7b). In both models, the induction algorithm mixes information from the previous token and stores it in a key-value format to predict the next token. The model retrieves this stored information to perform induction.

In Transformers, the previous token head attends to the preceding token, boosting the induction head's attention for next-token prediction through the OV circuit. In contrast, Mamba uses convolution in specific layers to merge neighboring tokens' information into the SSM state, which subsequent tokens query for induction (Figure 7).

## 6.3 LOCAL CONVOLUTION OF MAMBA PREFERS OFF-BY-ONE

A curious nuance is that Mamba merge information into the SSM input at token $B_1 + 1$ whilst Transformers directly performs token mixing between token $B_1$ and $A_2$. We term this the 'off by one' token mixing approach adopted by Mamba. It is theoretically possible that Mamba do not apply this strategy and instead write the information of $B_1$ into the states without *delay*. We also observe the same preference to 'off by one' in an early investigation of the IOI task (Wang et al., 2023) (Appendix H). We are still unclear why such mechanism forms or what it is for, which suggests further study in Mamba circuit analysis.

## 7 CONCLUSION

In this paper, we propose measuring cross-architecture universality with sparse autoencoder feature similarity. A novel metric is introduced to estimate the marginal effect of architectural difference to representational universality and find that most features are quite similar in both models. We also provided a intuitive interpretation of this metric: Pearson Correlation is affected by the semantic complexity of features. Thus we cancel out such effect by ablation. We also observe circuit-level analogy in induction mechanism and notice some interesting nuances called the *Off-by-One motif* between Mamba and Transformer.

## LIMITATIONS

**SAE Features as Universal Computational Units.** We are aware that SAEs, no matter how well they perform, yield only a subset of the *true* features (if exist) in the model. Though SAEs reveal levels of magnitude of matching features across models, we still lack a totally reliable way to locate those features uncaptured by SAEs. It is possible that these nuanced and rare features behave differently across models. It is also possible that SAEs capture a combination of the *true* features instead of the features themselves. Overall, Sparse Autoencoders and their implicit superposition hypothesis is still an immature research area. We think there are still a lot of room for improvement till we can measure the true universality of features.

**Qualitativeness of Circuit Universality.** Though we have shown that Mambas and Transformers implement the same induction algorithm, we have not yet quantitatively measured the similarity with some automated tools. It would be very interesting if there circuit discovery and comparison can be automated like some existing works in circuit discovery on Transformers (Conmy et al., 2023; Ge et al., 2024).

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

## A  SPARSE AUTOENCODERS TRAINING DETAILS

### A.1  COLLECTING ACTIVATIONS

We truncate each document to 1024 tokens and prepend a <bos> token to the beginning of each document. During training, we do not use the activation of <bos> and <eos> tokens.

It has been shown that the activations of different sequence positions of the same document are highly correlated and may lack diversity. To address this issue, it is common to introduce randomness into the training data. Our shuffling strategy is to maintain a buffer and shuffle it every time the buffer is refilled.

### A.2  INITIALIZATION AND OPTIMIZATION

The decoder columns $W_{:,i}^{dec}$ are initialized uniformly and normalized to have 2-norm of $\sqrt{\frac{2D}{F}} = \sqrt{\frac{2}{\text{expansion factor}}}$. We initialize the encoder weights $W^{enc}$ with the transpose of the decoder weights $W^{dec}$. The encoder bias $b^{enc}$ and decoder bias $b^{dec}$ are initialized to zero.

This ensures that the initialized SAE almost always achieve a near-zero reconstruction loss. It has been widely observed that this kind of initialization is beneficial for training SAEs.

We train SAEs with Adam optimizer with a $\beta_1 = 0.9$, $\beta_2 = 0.999$ of and $\epsilon = 10^{-8}$. The learning rate is set to to 8e-4 for all SAEs.

## B  DISTRIBUTION OF MAX PAIRWISE PEARSON CORRELATION ACROSS LAYER

We plotted the layer-wise distribution of MPPC calculated from Pythia to Mamba. For clarity, we divided the layers into four groups: 0–2, 3–5, 6–8, and 9–11, and presented the MPPC distribution histograms for each group (Figure 8).

As the layer number increases, the distribution gradually shifts to the left. In conjunction with the conclusion in Section 5.2 that as MPPC decreases, feature complexity tends to increase, we can infer that lower-layer features are simpler, while higher-layer features become increasingly complex.

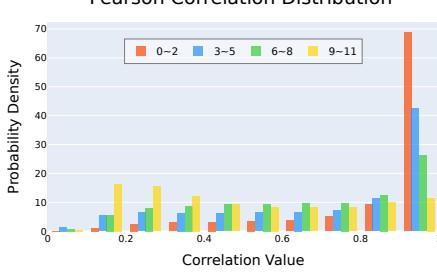

Figure 8: Histogram of the MPPC distribution for Pythia features, grouped by layers 0-2, 3-5, 6-8, and 9-11.

## C  BIDIRECTIONAL ANALYSIS OF MAX PAIRWISE PEARSON CORRELATION

The MPPC used in the main text is a one-way, irreversible metric, lacking symmetry. The experiments presented in the main text only report the results of finding the maximum Pearson correlation from Pythia to Mamba and Pythia (model seed variant). Here, we provide the reverse results, i.e., the distributions of $\rho_i^{p \to m}$ and $\rho_i^{m \to p}$, as well as $\rho_i^{p \to p'}$ and $\rho_i^{p' \to p}$ (Figure 9). Additionally, we show

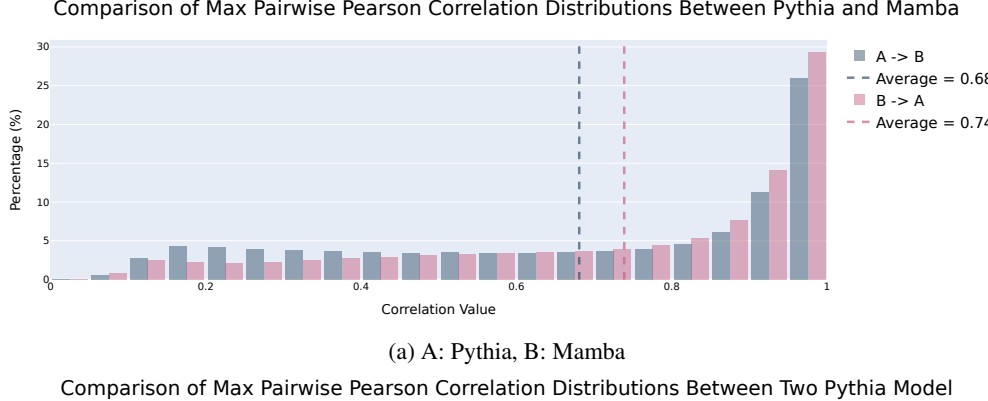

(a) A: Pythia, B: Mamba

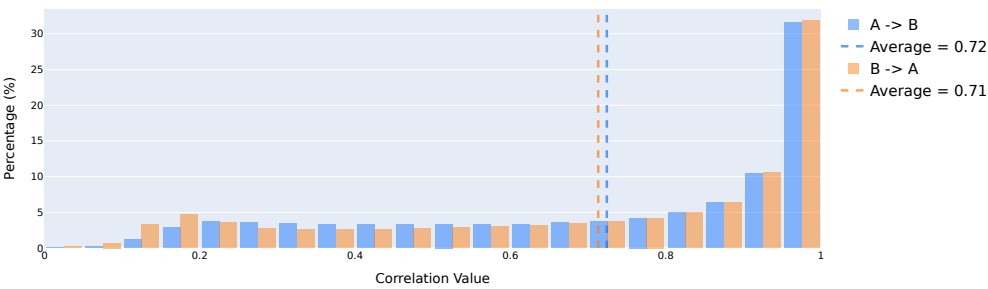

(b) A: Pythia, B: Pythia(model seed variant)

Figure 9: (a) MPPC distribution for Pythia and Mamba in both forward and reverse directions. (b) MPPC distribution for Pythia and Pythia (model seed variant) in both forward and reverse directions.

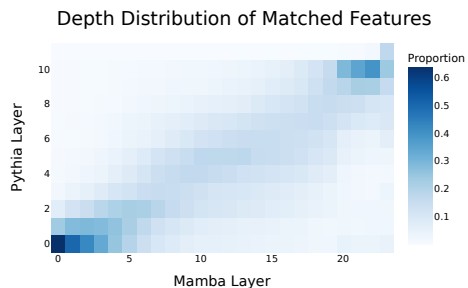

Figure 10: Frequency of the best matching feature pairs falling on each layer.

the Depth Distribution of Matched Features when calculating the MPPC from Mamba to Pythia (Figure 10).

The forward and reverse results for Pythia and Pythia (model seed variant) are very close. The reverse results for Pythia and Mamba outperform the forward results, primarily because Mamba has a higher proportion of lower-level features, which tend to have higher MPPC values (Section 5.2). This phenomenon is confirmed in the reverse Depth Distribution of Matched Features. The first five layers of Mamba tend to match with the first two layers of Pythia, especially the first layer, while the first seven layers of Mamba tend to match with the first three layers of Pythia—one more layer than proportional scaling would suggest. Lower-level features are generally simpler (Appendix B). Compared to Pythia, Mamba has a larger proportion of layers dominated by simpler features. However, depth specialization phenomenon is observed in both the forward and reverse matching processes.

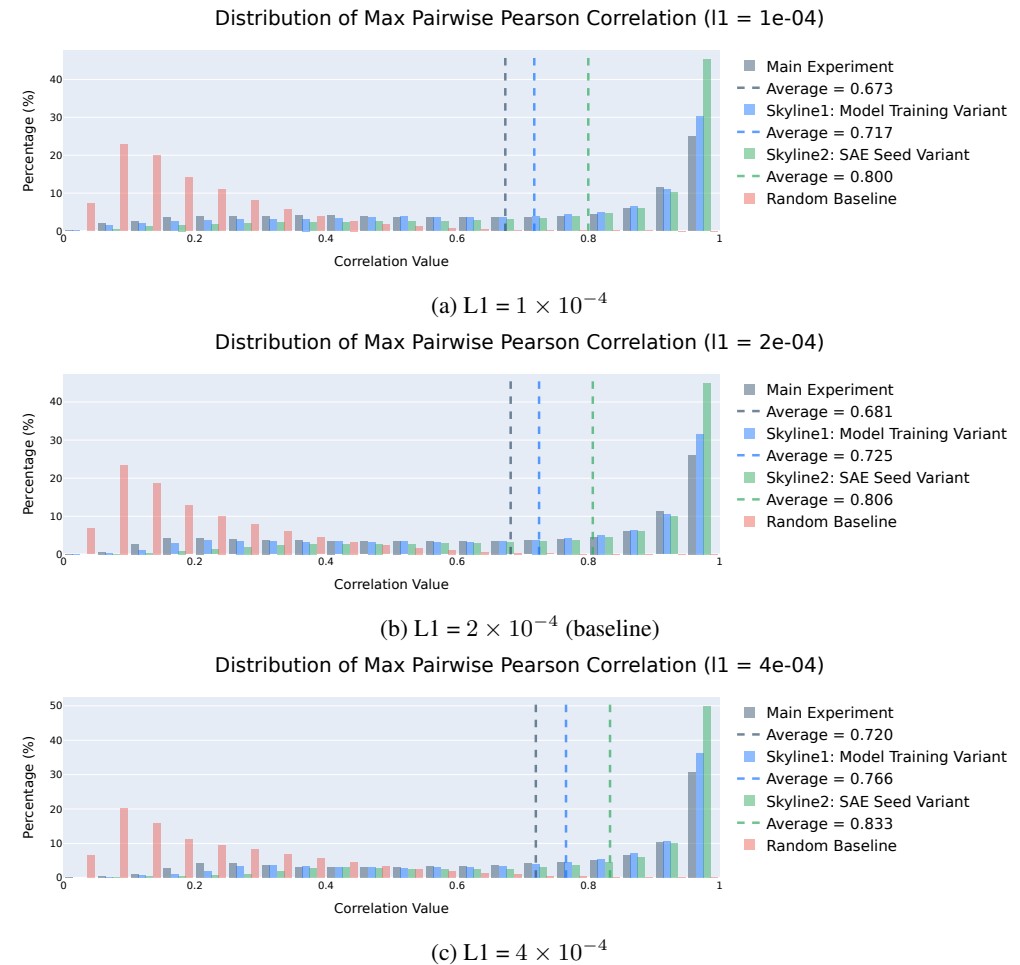

Figure 11: MPPC distributions under different L1 regularization strengths.

# D    COMPREHENSIVE ROBUSTNESS ANALYSIS OF MPPC ACROSS MULTIPLE DIMENSIONS

To thoroughly validate the robustness and generalizability of our method, we conducted extensive experiments across five key dimensions: hyperparameter sensitivity, dictionary size variation, dataset diversity, model architecture differences, and model scale. This comprehensive analysis ensures the reliability of our conclusions and demonstrates the method's stability under diverse conditions.

## D.1    HYPERPARAMETER SENSITIVITY ANALYSIS

**L1 Regularization Strength**    We systematically evaluated the impact of sparsity constraints by training SAEs with three distinct L1 coefficients: $1 \times 10^{-4}$, $2 \times 10^{-4}$ (baseline used in main experiments), and $4 \times 10^{-4}$. Figure 11 presents the resulting MPPC distributions, revealing consistent patterns across different regularization strengths.

## D.2    DICTIONARY SIZE VARIATION

To assess the impact of feature space dimensionality, we trained SAEs with expansion factors of 32 (baseline) and 64. The comparative MPPC distributions in Figure 12 demonstrate the method's stability across different dictionary sizes.

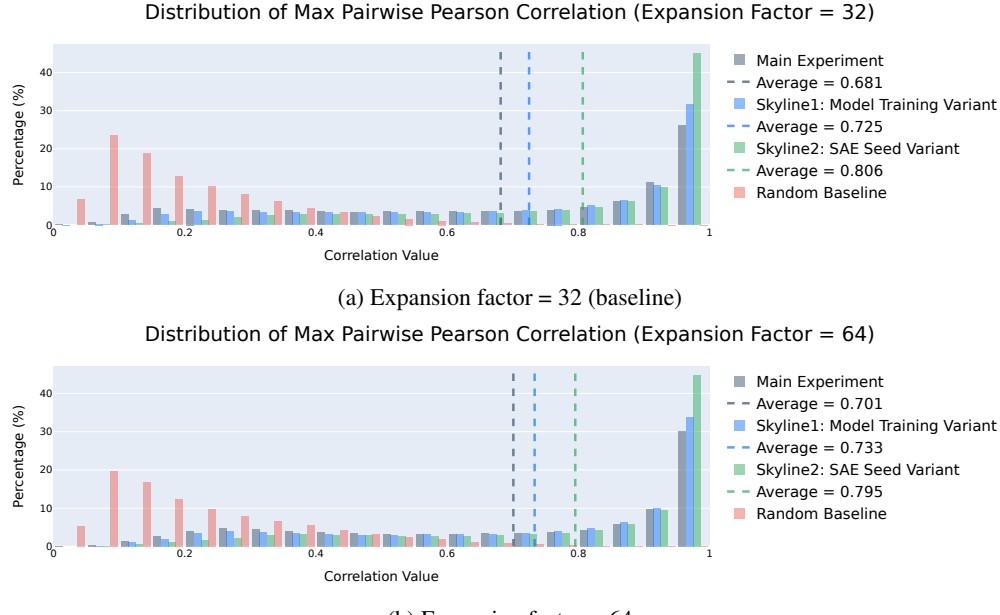

(a) Expansion factor = 32 (baseline)

(b) Expansion factor = 64

Figure 12: MPPC distributions with different dictionary sizes.

## D.3 DATASET ROBUSTNESS EVALUATION

We validated our approach on three distinct datasets:

- SlimPajama (baseline)
- RedPajama-Github (specialized code corpus)
- OpenWebText (general web text)

The consistent MPPC patterns across these datasets, as shown in Figure 13, highlight the method's robustness to data distribution variations.

## D.4 ARCHITECTURAL VARIANTS ANALYSIS

We extended our validation to different SAE architectures:

- Vanilla SAE (baseline)
- Top-k SAE (architecture variant)

Figure 14 illustrates the comparable performance across these architectural variants.

## D.5 MODEL SCALE ANALYSIS

To investigate scalability, we evaluated our method on larger models:

- Pythia-2.8B
- Pythia-2.8B-v0
- Mamba-2.8B

The results in Figure 15 demonstrate consistent behavior across different model scales.

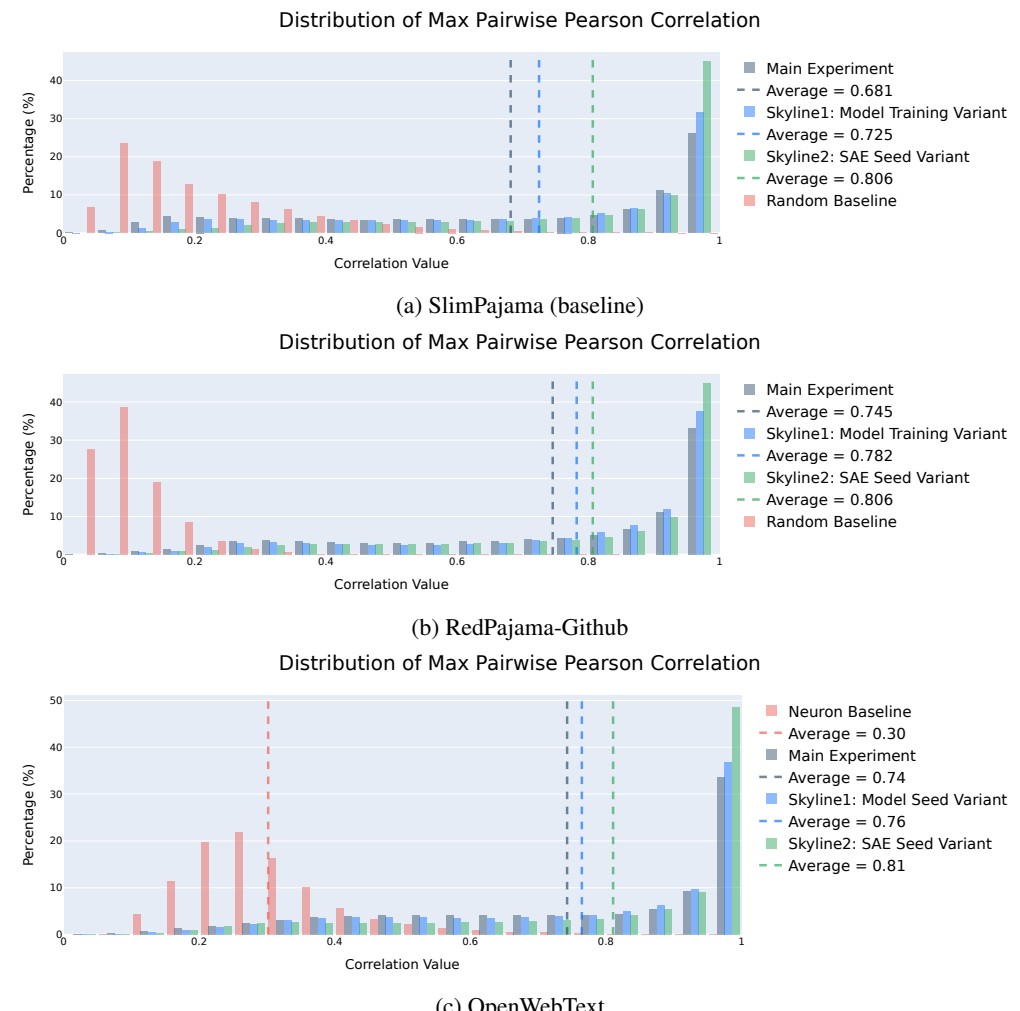

Figure 13: MPPC distributions across different training datasets.

## D.6 CROSS-ARCHITECTURE VALIDATION

Finally, we conducted a comprehensive cross-architecture analysis using three distinct model families:

- Pythia-160m (Transformer-based)
- Mamba-130m (State Space Model)
- RWKV-169m (RNN-Transformer hybrid)

The complete set of cross-architecture results in Figure 16 reveals consistent inter-model correlation patterns, further validating the generalizability of our approach.

## E  AUTOMATIC FEATURE SCORING PROMPTS

### E.1  USE FOR COMPLEXITY

We used the following prompt to have the LLM evaluate our Complexity:

> We are analyzing the activation levels of features in a neural network, where each feature activates certain tokens in a text. Each token's activation value indicates

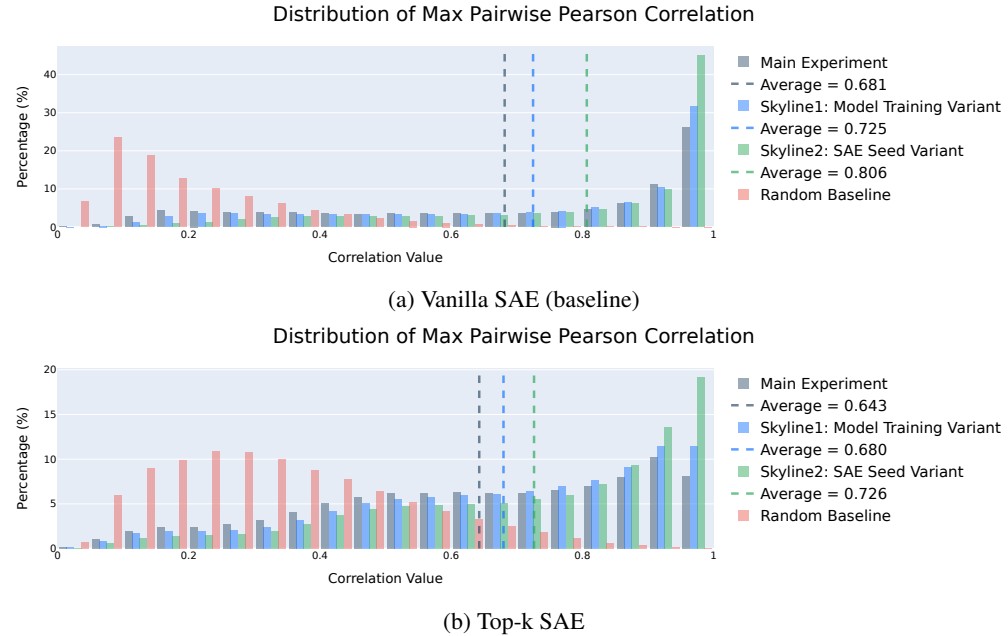

Figure 14: MPPC distributions for different SAE architectures.

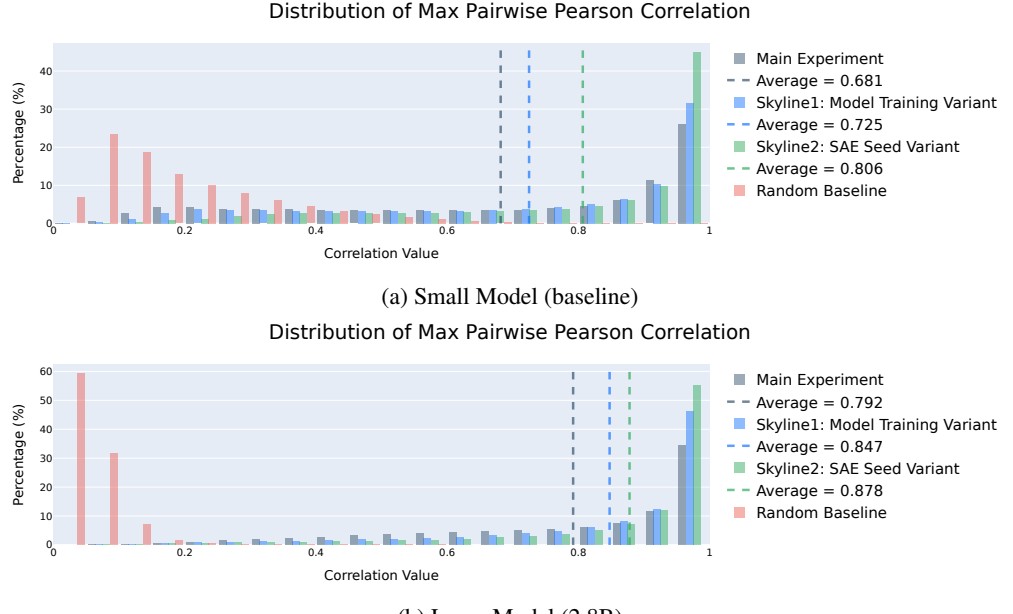

Figure 15: MPPC distributions across different model scales.

its relevance to the feature, with higher values showing stronger association. Your task is to infer the common characteristic that these tokens collectively suggest based on their activation values and give this feature a complexity score based on the following scoring criteria:

**Complexity**

- 5: Rich feature firing on diverse contexts with an interesting unifying theme, e.g. "feelings of togetherness"

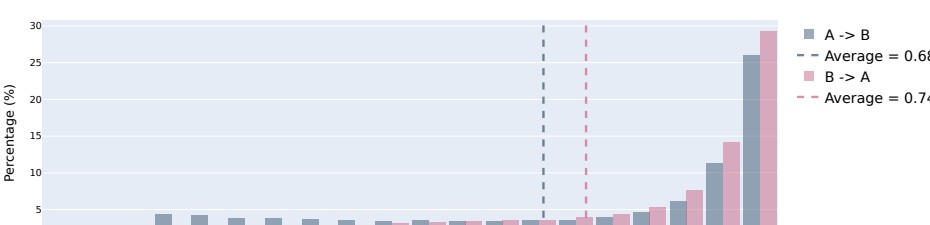

(a) Pythia vs. Mamba

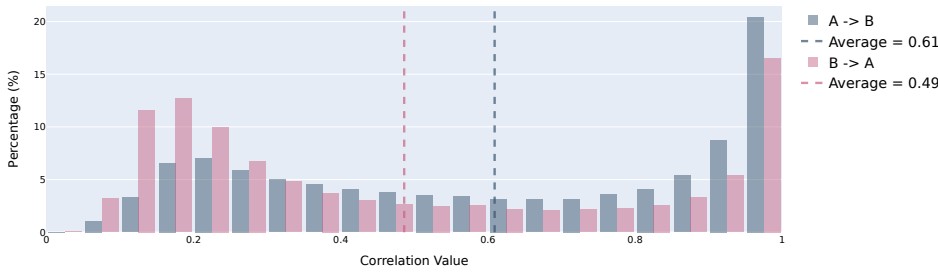

(b) Pythia vs. RWKV

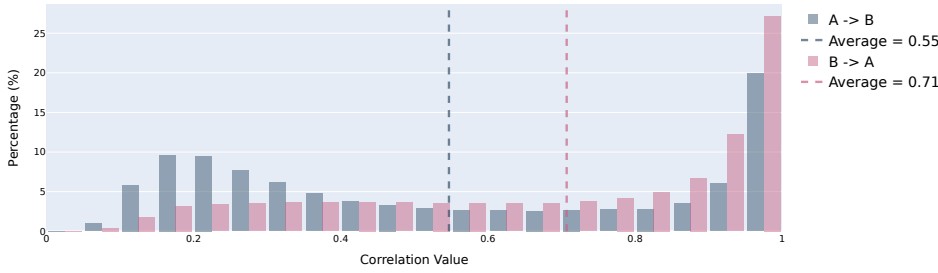

(c) RWKV vs. Mamba

Figure 16: Cross-architecture MPPC distributions.

- 4: Feature relating to high-level semantic structure, e.g. "return statements in code"
- 3: Moderate complexity, such as a phrase, category, or tracking sentence structure e.g. "website URLs"
- 2: Single word or token feature but including multiple languages or spelling, e.g. "mentions of dog"
- 1: Single token feature, e.g. "the token '('"

Consider the following activations for a feature in the neural network. Activation values are non-negative, with higher values indicating a stronger connection between the token and the feature. Don't list examples of words. You only need to give me a number! Just a number! It represents your score for feature complexity.

Following this prompt, we provided ten sentences containing top activations, each consisting of 17 tokens with the activation value for each token.

### E.2 USE FOR ACTIVATION CONSISTENCY

We used the following prompt to have the LLM evaluate our Activation Consistency:

We are analyzing the activation levels of features in a neural network, where each feature activates certain tokens in a text. Each token's activation value indicates its relevance to the feature, with higher values showing stronger association. Your task is to give this feature a monosemanticity score based on the following scoring criteria:

**Activation Consistency**

- 5: Clear pattern with no deviating examples
- 4: Clear pattern with one or two deviating examples
- 3: Clear overall pattern but quite a few examples not fitting that pattern
- 2: Broad consistent theme but lacking structure
- 1: No discernible pattern

Consider the following activations for a feature in the neural network. Activation values are non-negative, with higher values indicating a stronger connection between the token and the feature. You only need to give me a number! Just a number! It represents your score for feature monosemanticity.

Following this prompt, we provided ten sentences containing top activations, each consisting of 17 tokens with the activation value for each token.

## F CASE OF UNINTERPRETABLE FEATURES

We manually inspected a substantial number of features from the Pythia model with cross-architecture MPPC values below 0.2. We observed that these features are predominantly polysemantic, showing activations on text inputs with no discernible pattern or lacking clear structure. Figure 17 illustrates four randomly selected examples of uninterpretable feature pairs with different Pearson correlation values.

## G DATA RESULT OF PATH PATCHING

For each condition, we selected 128 randomly generated induction data points for path patching.

**Result1** The results of path patching for $h_{A_2-1}^{(l)}$ are shown in Table 1. Use [A][B']...[A] as corrupted input, [A][B]...[A] as clean input. Distance is the distance between two [A].

| distance \layer | 0 | 1 | 2 | 3 | 4 | 5 | 6 | 7 |
|---|---|---|---|---|---|---|---|---|
| 8 | 0.01 | -0.05 | 0.33 | -0.09 | 0.05 | 0.1 | -0.04 | 0.12 |
| 16 | 0.02 | -0.04 | -0.46 | -0.03 | -0.07 | 0.0 | 0.06 | 0.0 |
| 32 | -0.01 | 0.15 | 0.88 | 0.35 | -0.0 | 0.09 | 0.08 | 0.07 |
| 64 | -0.0 | -0.03 | -0.12 | 0.31 | -0.06 | 0.07 | 0.04 | 0.02 |
| 128 | 0.01 | -0.03 | -0.11 | -0.03 | 0.13 | 0.03 | 0.04 | -0.01 |
| distance \layer | 8 | 9 | 10 | 11 | 12 | 13 | 14 | 15 |
| 8 | 0.04 | 0.13 | 0.02 | 0.9 | 0.63 | -0.11 | 0.51 | -0.13 |
| 16 | -0.02 | 0.01 | 0.06 | -0.02 | -0.54 | 0.14 | -0.14 | 0.24 |
| 32 | 0.57 | -0.05 | 0.03 | 0.65 | 0.79 | -0.03 | 0.09 | 1.32 |
| 64 | 0.02 | 0.01 | 0.01 | -0.26 | -2.18 | 0.1 | 0.09 | 0.8 |
| 128 | -0.03 | 0.0 | -0.01 | 0.08 | -1.94 | 0.02 | -0.07 | 0.0 |
| distance \layer | 16 | 17 | 18 | 19 | 20 | 21 | 22 | 23 |
| 8 | 0.16 | -9.65 | 0.04 | -0.19 | -2.02 | -0.0 | -0.03 | -0.0 |
| 16 | -0.04 | -8.09 | -0.04 | -0.04 | -1.81 | -0.02 | -0.02 | -0.01 |
| 32 | 0.08 | -8.73 | 0.09 | -0.2 | -1.48 | -0.03 | -0.01 | -0.01 |
| 64 | 0.49 | -10.16 | -0.04 | -0.12 | -0.99 | -0.01 | -0.01 | -0.0 |
| 128 | -0.1 | -8.44 | -0.04 | 0.07 | -0.66 | -0.01 | -0.03 | -0.0 |

Table 1: The mean logit diff of token B after patching each state

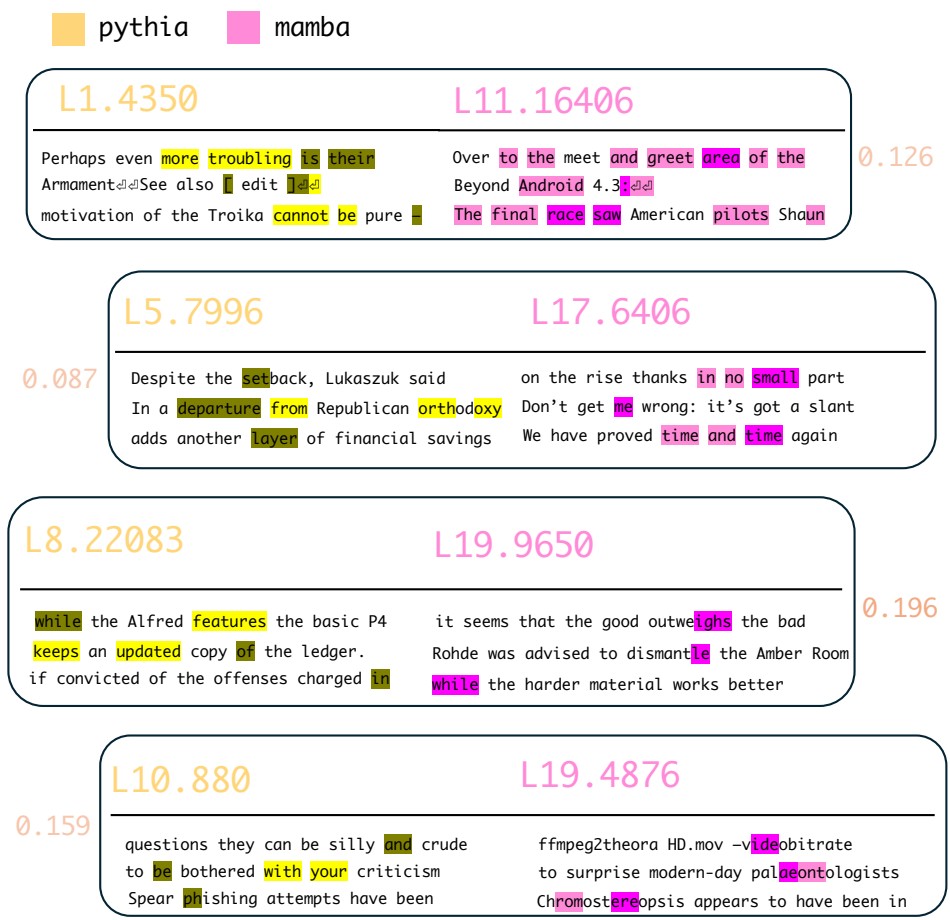

Figure 17: We present four uninterpretable cases of feature pairs with different Pearson correlation values. Each feature index is followed by activation examples shown below.

**Result2** The results of path patching for $c_i^{(17)}$ are shown in Table 2. Use [A][B']...[A] as corrupted input, [A][B]...[A] as clean input. Distance is the distance between two [A].

| distance \position | $A_1$ | $B_1$ | $B_1 + 1$ | $B_1 + 2$ | $B_1 + 3$ |
|---|---|---|---|---|---|
| 8 | 0.0 | -0.0 | -9.69 | 0.11 | -0.03 |
| 16 | 0.0 | -0.01 | -11.25 | 0.01 | 0.11 |
| 32 | 0.0 | -0.01 | -16.84 | -0.01 | -0.05 |
| 64 | 0.0 | -0.0 | -10.64 | -0.33 | 0.4 |
| 128 | 0.0 | -0.01 | -9.13 | 0.52 | 0.16 |

Table 2: The mean logit diff of token B after patching SSM input at each position

**Result3** The results of path patching for $x_i^{(17)}$ are shown in Table 4. Use [A][B']...[A] as corrupted input, [A][B]...[A] as clean input. Distance is the distance between two [A].

**Result4** The results of path patching for $x_i^{(17)}$ are shown in Table 4. Use [A'][B]...[A] as corrupted input, [A][B]...[A] as clean input. Distance is the distance between two [A].

| distance \position | $A_1$ | $B_1$ | $B_1 + 1$ |
|---|---|---|---|
| 8 | 0.0 | -9.11 | 0.26 |
| 16 | 0.0 | -15.17 | 0.24 |
| 32 | 0.0 | -13.95 | -0.06 |
| 64 | 0.0 | -13.71 | 0.27 |
| 128 | 0.0 | -9.5 | -0.11 |

Table 3: The mean logit diff of token B after patching $x_i^{(17)}$ at each position with [A][B']...[A] as corrupted input

| distance \position | $A_1$ | $B_1$ | $B_1 + 1$ |
|---|---|---|---|
| 8 | -8.27 | -0.88 | 0.04 |
| 16 | -9.44 | -0.52 | 0.17 |
| 32 | -11.35 | 0.72 | -0.02 |
| 64 | -9.07 | -1.5 | 0.2 |
| 128 | -8.96 | -0.54 | 0.13 |

Table 4: The mean logit diff of token B after patching $x_i^{(17)}$ at each position with [A'][B]...[A] as corrupted input

## H OFF BY ONE IN IOI TASK

### H.1 TASK DESCRIPTION

The indirect object identification (IOI) task involves sentences with two clauses: an initial clause (e.g., "When Mary and John went to the store") and a main clause (e.g., "John gave Mary a bottle of milk"). The initial clause introduces the indirect object (IO) and the subject (S), while the main clause mentions the subject again, indicating the action of the subject giving the object to the IO. The task is to predict the final token corresponding to the IO. We use "S1" and "S2" to represent the first and second occurrences of the subject, respectively.

### H.2 OCCURRENCE OF THE OFF-BY-ONE PHENOMENON

We use [IO][S]...[S]... as clean input and [IO'][S']...[S']... as corrupted input. Path patching is applied to $c_i^{(l)}$, allowing its influence on the logits through all paths except those involving other layer SSMs. We compute the logits difference between IO and S, as shown in Figure 18. It is observed that the $c$ causing the logit change always appears at the token immediately following IO or S. This suggests that the information from IO and S enters the SSM state at the next token, which is similar to the induction task phenomenon and represents another instance of the off-by-one effect.

## I HUMAN EXPERT EVALUATIONS AND CONSISTENCY WITH GPT-4 SCORING

In this section, we present a comprehensive human evaluation conducted to assess the complexity of 64 features and the monosemy of 32 features. The human experts were tasked with providing scores for each feature based on predefined criteria, ensuring a consistent and thorough assessment process. To gauge the alignment between human judgments and automated scoring, we compared the human scores with those generated by GPT-4. The comparison of the two sets of evaluations is illustrated in Figure 19. This analysis highlights the level of agreement between the expert and machine ratings.

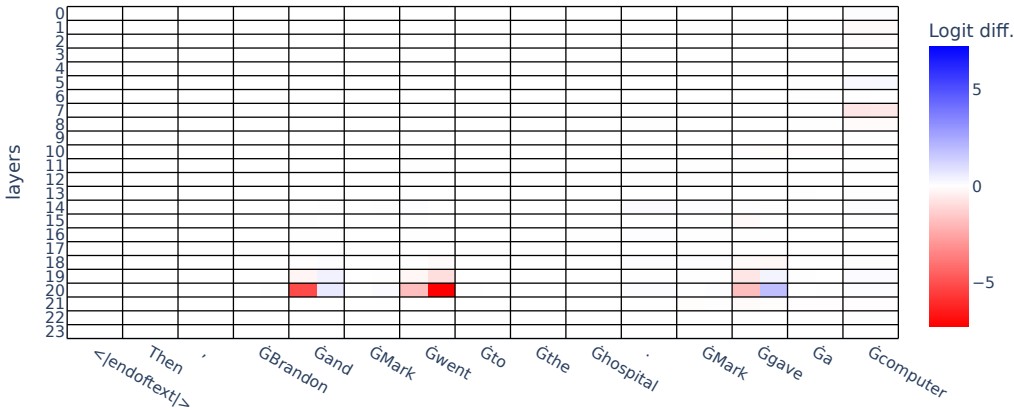

Figure 18: Logit changes after patching $c_i^{(l)}$ for each layer and each position. For each cell, the left side represents IO, and the right side represents S. Token positions are labeled using the tokens from the example sentence: "Then, Brandon and Mark went to the hospital. Mark gave a computer to Brandon."

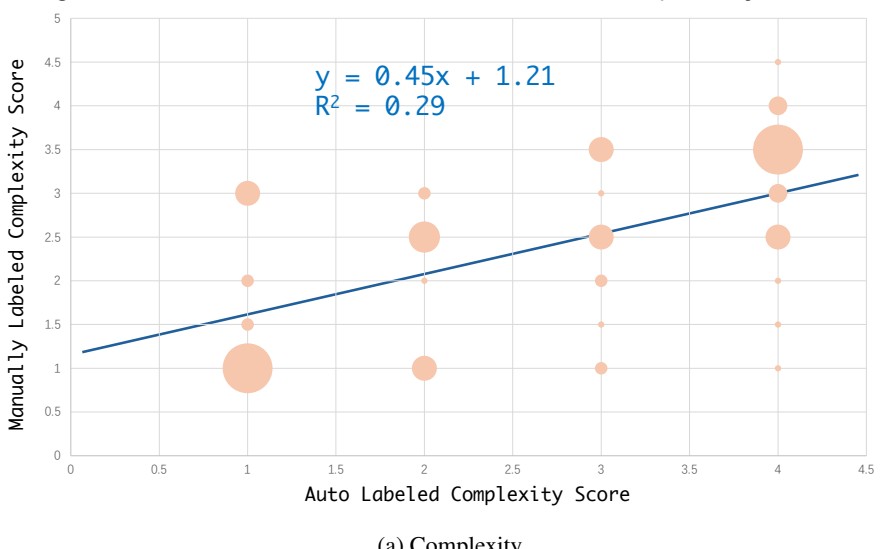

(a) Complexity

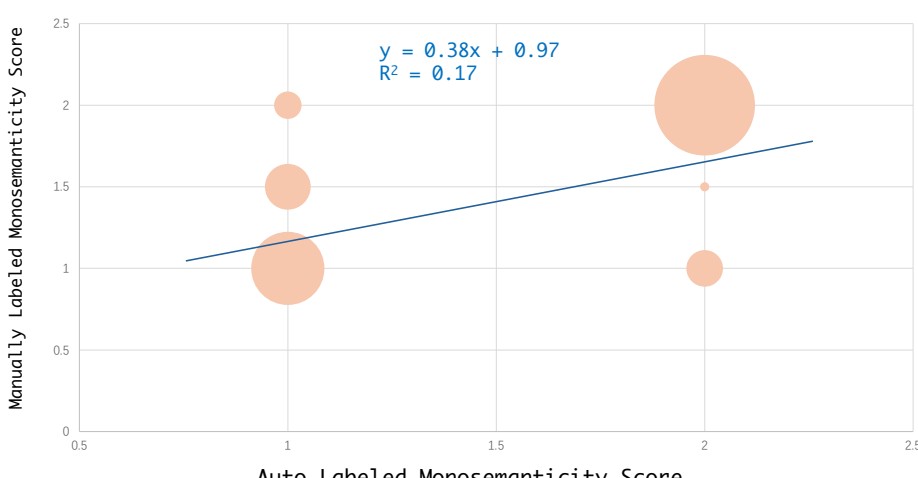

(b) Monosemanticity

Figure 19

