# OpenReview forum: "Towards Universality: Studying Mechanistic Similarity Across Language Model Architectures"
_ICLR.cc/2025/Conference — ICLR 2025 Poster_

### Official Review · Reviewer_jsst · 2024-10-29

**Soundness:** 3
**Presentation:** 3
**Contribution:** 2
**Rating:** 6
**Confidence:** 4

**Summary:**

This work investigates the similarity of sparse autoencoders (SAE) features and induction circuits between a Transformer-based and a comparable Mamba-based language model trained on the same dataset. Results show that many features from Transformers SAEs show a high max pairwise Pearson correlation with Mamba SAE features, with their depth along model layers roughly matching across the two models. The correlations found for cross-architecture comparison are compared to a neuron baseline and two skylines obtained from different models and SAE training seeds, showing that the cross-architecture comparison falls only slightly short of skylines. Authors further examine the correlation between cross-architectural matching features and their complexity, finding that features with the most overlap are generally simpler and more monosemantic. Finally, the authors briefly investigate the similarity between induction circuits in the same architectures using path patching, finding a similar mechanism mediated by the convolutional operation of the Mamba architecture. Notably, the information mixing necessary for the induction operation is performed earlier in Mamba ("off-by-one" mixing).

**Strengths:**

The universality evaluation pursued in this paper is timely and relevant, given recent advances in non-Transformer architectures for language modeling. The baseline and skylines employed in this work are well-motivated and provide helpful reference points for the analysis. The analysis of feature correlation based on complexity is also interesting, showing convincing proof that most commonalities across architectures are found for simpler SAE features. Overall, the figures are designed clearly and compellingly to support the findings detailed in the main body of the paper.

**Weaknesses:**

**Novelty of the findings** While to my knowledge, this is the first work evaluating the similarity of SAE features between Transformers and Mamba models, other works such as [Paulo et al. 2024](https://arxiv.org/pdf/2404.05971) and [Sharma et al. 2024](https://arxiv.org/abs/2404.03646) already showed that many interpretability approaches such as logit lenses, steering vectors, and probes produce similar results across a variety of LLM architectures (Mamba, Transformer and RWKV). It is, hence, not particularly surprising that such findings extend to SAE decompositions of model activations.

**Generality of findings for larger models**  Authors experiment only with 2 tiny models with ~100M parameters each. This can be reasonable in light of the requirements (same training data, effort of training SAEs on each layer), but these systems are significantly smaller than those used by [Paulo et al. 2024](https://arxiv.org/pdf/2404.05971) for comparable cross-architecture interpretability experiments. Notably, larger checkpoints for both models used by the authors are publicly available, including the same training data for control, and could have been used to further prove the generality of the reported results. Importantly, without further experiments, it cannot be excluded that the limited capacity of tiny models might be the main motivation behind the high similarity features and circuits across the two architectures, and this could not be the case for more capable models with e.g. 1B or 7B parameters.

**Multiple Comparison and Correlation Analysis without Hypothesis Testing** The maximal correlation of feature activation patterns with other (24576 x # of layers) features is bound to be quite high due to the enormous amounts of comparisons. In Section 4.4, no hypothesis is formulated regarding the expected similarity of features found across the four tested variants, and consequently, no significance measure for the correlation coefficients is reported. As a result, conclusions regarding the similarity of Mamba and Pythia SAE features are ambiguous (e.g. the statement "[...] our neuron baseline almost exhibits zero sign of similarity between Mamba and Pythia" at line 268 does not agree with Figure 3a, where at least 15% of neurons exhibit a correlation > 0.4). To make the analysis more convincing, a clear hypothesis regarding the degree of similarity in resulting SAE features should have been formulated and tested for baseline, experiment, and skylines, each including a correction procedure such as the Bonferroni method to account for multiple comparisons.

**Minor formatting/clarification points:**

Line 135: The mention "$F_a$ and $F_b$ are some kinds of operation function." is too generic in this context. The purpose of these functions should be specified, and at least one example of functions used for this purpose should be provided.

Line 179: Broken reference.

Figure 1 is too tight with the inline text, making the distinction between caption and main body text unclear.

Section 4.1: title is ungrammatical. I imagine you meant something like "Searching for / In search of Interpretable Primitives".

Line 202: Clarify that you mean all features for all SAEs across all model layers (it becomes clear only from Figure 3c later in the paper)

Line 263: The acronym MPPC is never introduced alongside its meaning.

Figure 5: The mention "both Model Seed Variant and Cross-Arch SAE MPPC exhibit correlation, while one in SAE Seed Variant is weaker" in the caption is not very meaningful, since the trends for all three variants are pretty similar. For part (b), the mention "scores ranging from 1 (No) to 2 (Yes)" is confusing: it would be better to say "Distribution of MPCC for polysemantic (1) and monosemantic (2) auto-generated feature labels."

**Questions:**

What justified the choice of plain SAEs over more performant variants such as [Gated](https://arxiv.org/abs/2404.16014) or [Top-K SAEs](https://openai.com/index/extracting-concepts-from-gpt-4/)? It is currently hard to gauge the impact of this choice on the obtained results, and whether findings could have been different if improved SAE variants were tested.

---

> ### Author Response · Authors · 2024-11-23
> **Response to Reviewer jsst (1 / 5)**
>
> We sincerely appreciate your recognition of our work, with highlights in motivation, interest, soundness and clarity. We would also like to acknowledge the thorough and constructive feedback and questions provided which help strengthen our work, which we summarize as follows:
> - Novelty of the findings;
> - Generality of findings for larger models;
> - Statistical significance tests;
> - Choice of SAE variants.
>
> All of these aspects are helpful and insightful. In addition, thanks for pointing out the typos and clarification points in our submitted manuscript, which we have all fixed in our updated version.

---

> ### Author Response · Authors · 2024-11-23
> **Novelty of the Findings (2 / 5)**
>
> >While to my knowledge, this is the first work evaluating the similarity of SAE features between Transformers and Mamba models, other works such as Paulo et al. 2024 and Sharma et al. 2024 already showed that many interpretability approaches such as logit lenses, steering vectors, and probes produce similar results across a variety of LLM architectures (Mamba, Transformer and RWKV). It is, hence, not particularly surprising that such findings extend to SAE decompositions of model activations.
>
> Thanks for pointing this out. [Paulo et al. 2024](https://arxiv.org/pdf/2404.05971) is a highly relevant work which we did not cite in our submitted manuscript. We have included this in our new revision. Apologies for this.
>
> We would like to point out that **although our work mainly falls on the line of transferrability of interpretability methods across LM architectures**, which has already been explored by existing literature, **our findings provide new insights into the following aspects**:
> - The analysis of depth specialization of SAE features across models suggests **the existence of an architecture-agnostic feature hierarchy for language modeling**.
> - The induction circuit similarity analysis, though being rather simple, serves as a supporting example of **architecture-agnostic circuit motif for language modeling**.
> - **Our complexity viewpoint of MPPC raises a broad concern for cross-architecture model diffing**. For example, a recent promising model diffing method [crosscoder](https://transformer-circuits.pub/2024/crosscoders/index.html) is also prone to false positive of divergent features, where features activating on the same concept might be identified as different ones due to their respective preference for specific instances. This suggests the need for sanity checks of subsequent model diffing methods.
>
> We would like to appreciate it again for pointing out a missing related work. And we are glad to have further discussion on the novelty of our findings.

---

> ### Author Response · Authors · 2024-11-23
> **Generalization to Larger Models (3 / 5)**
>
> >(Reviewer 4GEc) How do you expect your findings to scale to larger models? Did you observe whether model size impacts universality between architectures? Could smaller or larger versions of Transformers and Mambas exhibit different degrees of feature similarity?
>
> >(Reviewer nuP5) (Minor) The size of LMs is limited. Only ~100m models are used in the experiments.
>
> >(Reviewer jsst) Without further experiments, it cannot be excluded that the limited capacity of tiny models might be the main motivation behind the high similarity features and circuits across the two architectures, and this could not be the case for more capable models with e.g. 1B or 7B parameters.
>
> Thanks for pointing this out, which greatly helps improve our work.
>
> We additionally conduct the experiment for **2.8B variants of both models**, giving us the following results:
> | Mean MPPC / Model Size       | 130M (original) | 2.8B  |
> |------------------------------|-----------------|-------|
> | Main experiment              | 0.681           | 0.792 |
> | Skyline 1 (Model Training Variant) | *0.725*           | *0.847* |
> | Skyline 2 (SAE Seed Variant)   | **0.806**           | **0.878** |
>
> **We have incorporated these updates in Sections 4.4 and Appendix D.4** of the revised manuscript for further clarity and elaboration.

---

> ### Author Response · Authors · 2024-11-23
> **Statistical Significance Tests (4 / 5)**
>
> >(Reviewer 4GEc) Have you performed any statistical significance tests to support your claims of feature similarity and universality?
>
> >(Reviewer jsst) In Section 4.4, no hypothesis is formulated regarding the expected similarity of features found across the four tested variants, and consequently, no significance measure for the correlation coefficients is reported.
>
> This is an important part we missed in our submitted manuscript and we are very thankful for pointing this out.
>
> We additionally establish a **random baseline** by calculating MPPC for each Pythia feature against a random SAE **with matching feature count and sparsity level** (by masking all but the TopK activating features) to the Mamba SAE to observe the impact of feature quantity and sparsity on MPPC distribution.
>
> We denote mean MPPC as the random variable \\( x \\). To evaluate whether the result of our main experiment (\\( x = 0.681 \\)) belongs to a distribution with the same mean as the random baseline, we conducted a **Hypothesis Testing**. Specifically, we tested the null hypothesis (\\( H_0 \\)) that **the mean of the experimental group is equal to the mean of the random baseline group** (\\( \mu_{\text{experiment}} = \mu_{\text{baseline}} \\)) against the alternative hypothesis (\\( H_1 \\)) that the two means are different (\\( \mu_{\text{experiment}} \neq \mu_{\text{baseline}} \\)).
>
> The random baseline group consisted of 16 samples(repeated the baseline 16 times and calculate mean MPPC for each) with a mean of \\( 0.1944 \\) and a standard deviation of \\( 0.00046 \\). The experimental group consisted of a single sample with \\( x = 0.681 \\). Since the sample size of the experimental group is one, we applied a two-sample \\( t \\)-test using the pooled standard deviation from the baseline group. Yielding a \\( t \\)-value of \\( -1026.24 \\). The degrees of freedom are \\( n_{\text{baseline}} + n_{\text{experiment}} - 2 = 15 \\). Using a **one-tailed \\( t \\)-test**, the resulting \\( p \\)-value is:
>
>
> $p = 4.54 \times 10^{-38}$
>
> Given that the \\( p \\)-value is significantly smaller than any conventional significance level (\\( \alpha = 0.05 \\)), we **reject the null hypothesis**. This indicates that the experimental result (\\( x = 0.681 \\)) is highly unlikely to belong to a distribution with the same mean as the random baseline.
>
> **We have incorporated these updates in Sections 4.1 and 4.4 of the revised manuscript** for further clarity and elaboration.

---

> ### Author Response · Authors · 2024-11-23
> **Choice of SAE Variants (5 / 5)**
>
> >What justified the choice of plain SAEs over more performant variants such as Gated or Top-K SAEs? It is currently hard to gauge the impact of this choice on the obtained results, and whether findings could have been different if improved SAE variants were tested.
>
> Thanks for this valuable question. Vanilla SAEs are indeed less performant and somewhat outdated in a retrospective view. We conduct the main experiment and both skylines with TopK SAEs and get the following results:
>
> | Mean MPPC / SAE Variant         | Vanilla (Original) | TopK   |
> |---------------------------------|--------------------|--------|
> | Main experiment                 | 0.681              | 0.643  |
> | Skyline 1 (Model Seed Variant)  | *0.725*              | *0.680*  |
> | Skyline 2 (SAE Seed Variant)    | **0.806**              | **0.726**  |
>
> **We have incorporated these updates in Sections 4.4 and Appendix D.3 of the revised manuscript** for further clarity and elaboration.

---

> > ### Comment · Reviewer_jsst · 2024-11-25
> > **Reply to Rebuttal**
> >
> > Thank you for your reply and for running these further experiments, which indicate that the results you observed seem to hold for different model sizes and SAE architectures. Your comment regarding the novelty of your findings looks reasonable. However, I would argue that being able to transfer steering and vocabulary projection approaches across model architectures might already be a valid indicator of similar latent feature spaces. Still, I think your comments are generally valid and should be incorporated in the paper's discussion section (especially the part concerning the findings' applicability to model diffing).
> >
> > In light of this, I am willing to slightly raise my score to reflect the improved generality of the findings. However, I remain skeptical about the statistical testing you performed. My main concerns are:
> > - Using a random SAE as a null hypothesis is a very low bar to claim feature similarity between the two models. Such tests may, at best, confirm that the similarity between latents is higher than random chance, which is unsurprising given that both models were trained on the same data. An evaluation comparing the similarity of features cross-layer vs. cross-architecture would have been more interesting in this regard (e.g., Is the similarity of features from upper layers of Mamba and upper layers of Pythia higher than between upper and lower layers in the same model?)
> > - I am confused about what the samples in your testing correspond to since I thought 0.681 was the averaged MPCC over a set of samples rather than the max SAE feature correlation for a single sample. Why wasn't the correlation measured over the full set of samples for which the experiment MPCC was computed? With n=1, you cannot estimate the variance of the experimental group, and since the t-test you are conducting is designed to account for uncertainty in both sample means, using the pooled standard deviation from the baseline group does not capture the variation around the mean for the experimental group.

---

> > > ### Author Response · Authors · 2024-11-25
> > > **Response to Reviewer jsst Comments (2 / 2)**
> > >
> > > >I am confused about what the samples in your testing correspond to since I thought 0.681 was the averaged MPCC over a set of samples rather than the max SAE feature correlation for a single sample. Why wasn't the correlation measured over the full set of samples for which the experiment MPCC was computed? With n=1, you cannot estimate the variance of the experimental group, and since the t-test you are conducting is designed to account for uncertainty in both sample means, using the pooled standard deviation from the baseline group does not capture the variation around the mean for the experimental group.
> > >
> > > Thank you very much for raising these important concerns.
> > >
> > > With regard to your question on "Why wasn't the correlation measured over the full set of samples", we realize that **our explanation may not have been sufficiently clear**. Specifically, when we stated that the "random baseline group consisted of 16 samples", we intended to convey that **we repeated the baseline 16 times and calculated the mean MPPC for each repetition**. To address this, we have revised our response above to ensure a more detailed and precise explanation.
> > >
> > > Concerning your observation regarding the variance of the experimental group, **we acknowledge that we made the simplifying assumption that the variance of the experimental group is the same as that of the baseline group**. While we recognize that this assumption may not hold perfectly, we believe **it has minimal impact on the overall conclusion**. This is because the mean of the experimental group is significantly larger than that of the baseline group, which allows us to conclude that the means of the two distributions are distinct, even under the possibility of slightly larger variances.
> > >
> > > We sincerely regret any confusion caused by the lack of clarity in our initial explanation and appreciate your thorough review and constructive feedback. Your meticulous and rigorous approach has been invaluable, and we hold the utmost respect for your careful evaluation of our work.

---

> ### Author Response · Authors · 2024-11-25
> **Response to Reviewer jsst Comments (1 / 2)**
>
> >Using a random SAE as a null hypothesis is a very low bar to claim feature similarity between the two models. Such tests may, at best, confirm that the similarity between latents is higher than random chance, which is unsurprising given that both models were trained on the same data. An evaluation comparing the similarity of features cross-layer vs. cross-architecture would have been more interesting in this regard (e.g., Is the similarity of features from upper layers of Mamba and upper layers of Pythia higher than between upper and lower layers in the same model?)
>
> Thank you very much for your thoughtful feedback and for adjusting the rating. We greatly appreciate the time and effort you have invested in reviewing our paper.
>
> We understand your concern, which, if we have interpreted correctly, points to the baseline being relatively weak. You note that **the substantial performance of the main experiment over the baseline might not sufficiently reflect the extent of cross-model feature similarity**.
>
> To address this, we would like to clarify that **our paper used the relatively small difference between the main experiment and Skyline1 to reflect this similarity**. Specifically, we highlighted that more than 60% of features exhibit near-zero MPPC differences (absolute value < 0.05). This implies that, for the majority of features (and their corresponding semantics), the similarity between Mamba and Pythia is almost indistinguishable from the similarity between Pythia and models of the same architecture.
>
> That being said, your suggestion is particularly insightful. **Comparing cross-layer similarities to cross-architecture similarities could indeed provide a new and meaningful baseline** for evaluating cross-model similarity. Inspired by your comment, we have conducted additional experiments to explore this idea. Specifically, **we calculated the similarity between high layer(12\~23) features and lower layer(0\~11) features within Mamba, as well as the similarity between high layer features of Mamba(12\~23) and Pythia(6\~11)**. In the table below, the column headers represent the range of MPPC values calculated by mamba(all layers) to pythia(all layers), where **lower values indicate relatively higher feature complexity** (which sometimes may simply result from the absence of matching semantically similar features). The results are as follows:
>
> | Comparison Object \ MPPC Interval       | 0.2~0.3 | 0.3~0.4 | 0.4~0.5 | 0.5~0.6 | 0.6~0.7 | 0.7~0.8 | 0.8~0.9 | 0.9~1.0 |
> |---------------------------|----------|----------|----------|----------|----------|----------|----------|----------|
> | mamba high -> pythia high | 0.239    | 0.338    | **0.436**    | **0.531**    | **0.622**    | 0.707    | 0.785    | 0.873    |
> | mamba high -> mamba low   | **0.264**    | **0.361**    | **0.436**    | 0.521    | 0.615    | **0.732**    | **0.863**    | **0.961**    |
>
> It can be observed that for **relatively simple features** (the rightmost three columns), Mamba's higher layers and lower layers exhibit stronger similarity. For **relatively complex features** (the third, fourth, and fifth columns), Mamba's higher layers and Pythia's higher layers show stronger similarity(same in the third column). As for the leftmost two columns, we speculate that this might be due to some Mamba features **failing to find matches** in Pythia.
>
> We hope this additional analysis addresses your concern and provides further clarity on the cross-model feature similarity presented in our paper.
>
> Thank you once again for your valuable insights, which have helped us improve the rigor and depth of our study. Please let us know if you have any further suggestions or questions.

---

### Official Review · Reviewer_eqTo · 2024-11-04

**Soundness:** 4
**Presentation:** 3
**Contribution:** 3
**Rating:** 8
**Confidence:** 3

**Summary:**

This paper presents an exploration of the Transformer and Mamba models through mechanistic interpretability methodology. Despite the architectures being very different, the features and circuits of these models turn out to be very similar.

**Strengths:**

- The paper is clearly written.
- I appreciate the idea of using skylines, as it helps support the authors' claims.
- The results are interesting and useful for further research.

**Weaknesses:**

I couldn't identify any specific weaknesses. However, below are some suggestions that could enhance this work from my perspective:


- It would be interesting to explore more circuits through SAE, as suggested in [1] (see Section 4.3). However, it is unclear where SAE should be placed within the Mamba architecture to achieve similar features.
- While the Pearson correlation appears to be a natural choice for measuring feature similarity, it assumes that the feature space has linear properties. It might be worthwhile to explore other correlation measures, such as distance correlation, which could potentially yield better results.
- A clear statement clarifying that MPPC refers to the maximum Pearson correlation between models' features is needed to improve understanding.

[1] Interpreting Attention Layer Outputs with Sparse Autoencoders (Kissane et al.)

**Questions:**

- While the heatmap matrix in Figure 3c is mainly diagonal, I can see that there is a cluster of features located in the last layer of the Pythia and distributed fairly uniformly in the middle layers of Mamba. Can the authors clarify the meanings of these features?

---

> ### Author Response · Authors · 2024-11-23
> **Response to Reviewer eqTo (1 / 4)**
>
> We sincerely appreciate your recognition of our work, with highlights in novelty, clarity, and impact. We would also like to acknowledge the thorough and constructive feedback and questions provided which help strengthen our work, which we summarize as follows:
> - Exploring more circuits with SAEs;
> - Exploring more correlation metrics;
> - Clarify the implication of the last Pythia layer exception in depth-specificity analysis.
>
> All of these aspects are helpful and insightful. And many thanks for suggestions to clarify the meaning of the acronym MPPC, which we have included in our updated version.

---

> ### Author Response · Authors · 2024-11-23
> **Exploring More Circuits with SAEs (2 / 4)**
>
> >It would be interesting to explore more circuits through SAE, as suggested in [1] (see Section 4.3).
>
> Thanks for this suggestion. We think that SAE feature circuits, despite some existing literature investigating this problem[1, 2, 3], is still not what we think a mature method to help support our claims due to its time and computation complexity. In addition, **we expect findings of SAE feature circuits to be a finer-grained extension of head-level or block-level circuit analysis**, which may not contradict with our findings.
>
> [1] [Dictionary Learning Improves Patch-Free Circuit Discovery in Mechanistic Interpretability: A Case Study on Othello-GPT](https://arxiv.org/abs/2402.12201)
>
> [2] [Sparse Feature Circuits: Discovering and Editing Interpretable Causal Graphs in Language Models](https://arxiv.org/abs/2403.19647v1)
>
> [3] [Transcoders Find Interpretable LLM Feature Circuits](https://arxiv.org/abs/2406.11944v1)

---

> ### Author Response · Authors · 2024-11-23
> **Exploring More Correlation Metrics (3 / 4)**
>
> >It might be worthwhile to explore other correlation measures, such as distance correlation, which could potentially yield better results.
>
> Thank you so much for your insightful suggestion! We genuinely appreciate the thoughtfulness behind it. In our paper, we used Pearson Correlation as a metric to measure feature similarity. However, we agree that, influenced by factors such as feature complexity, Pearson Correlation may not fully capture the semantic similarity of features. **Exploring more sophisticated and semantically aligned similarity metrics is indeed a promising direction for future research**.
>
> That being said, Distance Correlation, while conceptually compelling, **presents significant computational challenges**. **Its time and space complexity scale as $O(n^2)$ with the number of sampled tokens \\(n\\), compared to the much more efficient \\(O(n)\\) complexity of Pearson Correlation**. Given the resource constraints we faced, we were unable to employ Distance Correlation in this work. Nonetheless, we see great potential in this idea and are excited to explore it in our future research endeavors.
>
> Once again, thank you for your valuable feedback—it has given us fresh perspectives and inspired directions for further improvement!

---

> ### Author Response · Authors · 2024-11-23
> **Implication of the Last Pythia Layer Exception in Depth-Specificity Analysis (4 / 4)**
>
> >While the heatmap matrix in Figure 3c is mainly diagonal, I can see that there is a cluster of features located in the last layer of the Pythia and distributed fairly uniformly in the middle layers of Mamba. Can the authors clarify the meanings of these features?
>
> Thanks for pointing this out. Apologies for not explaining this exception which is indeed confusing without further details. We expect this to be reasonable for the following reasons:
> - The residual stream after the last layer is directly connected with the unembedding (interleaved by a LayerNorm), so it should **most contain information about next token prediction**, rather than that about past tokens, making it substantially different from lower-layer residual stream activations.
> - **One piece of evidence of this is the ultra-high norm of the last layer residual stream**, which was reported in the third figure in [this post](https://www.alignmentforum.org/posts/8mizBCm3dyc432nK8/residual-stream-norms-grow-exponentially-over-the-forward). One can think of this as "Overwriting the whole residual stream to focus on predicting".
> - For example, in [this last-layer residual stream SAE](https://www.neuronpedia.org/gpt2-small/11-res_post_32k-oai), **the interpretation for most features are only clear if one looks into the top logits they contribute to**, which is much less often the case for lower-layer ones.

---

> > ### Comment · Reviewer_eqTo · 2024-11-26
> >
> > Thank you for your responses. My score is already 8, and I believe this paper is useful and interesting for the MI community. Good luck in the discussion period.

---

> > > ### Author Response · Authors · 2024-11-26
> > > **Response to Reviewer eqTo Comments**
> > >
> > > Thank you very much for your thoughtful comments and feedback. We truly appreciate your kind words about the usefulness and interest of our paper for the MI community. We are grateful for your time and effort in reviewing our work, and we are pleased to hear that you find it valuable.
> > >
> > > We also appreciate your score of 8 and will continue to refine the paper during the discussion period based on the feedback we receive.

---

### Official Review · Reviewer_nuP5 · 2024-11-04

**Soundness:** 3
**Presentation:** 3
**Contribution:** 3
**Rating:** 6
**Confidence:** 3

**Summary:**

This paper studies the mechanistic similarity between language model structures (Mamba, RWKV, Transformer). The authors focus on their Universality, a hypothesized property that suggests different neural architectures implementing similar algorithms on similar tasks.

In the first part of the experiment section, they use Sparse Autoencoder (SAE) as the major tool for their analysis. The representations from two LM architectures are taken to train SAEs. The latent vectors in the SAEs, in which each dimension corresponds to various syntactic and semantic phenomena, exhibit mechanical similarities, and it is possible to find a matching between the vectors from different LM architectures.

In the second part, the authors study the induction behavior of two LM architectures. They found that the 17th layer of Mamba is the most important for the LM’s inductive ability.

I think this paper studies an important problem and is well executed. I found the experiments in this paper to be well implemented. My only concern is with the role of circuit analysis experiments. They are indeed very interesting but I’m not sure how they contribute to building a mechanistic analogy between SSMs and Transformers. Do Transformers have such layer-specific behavior when it comes to inductive ability? Is there a way to empirically verify the claims in Sec 6.2?

Minor:

Appendix D: How does the feature mapping between RWKV and Mamba look like?
Sec 6.1: Is the layer-17 phenomenon robust to random initializations? I.e., if one retrains the SSM with another seed, would layer 17 still be the key in induction?
Line 179: missing section reference.
Line 861: missing space between ‘Universality’ and ‘is’

**Strengths:**

* This paper studies an important problem.
* It makes good use of sparse autoencoders for analysis.
* The experiments in this paper are well implemented.

**Weaknesses:**

* The role of circuit analysis experiments is unclear.
* The claims made in Section 6.2 need to be empirically supported.
* (Minor) The size of LMs is limited. Only ~100m models are used in the experiments.

**Questions:**

* Do Transformers have such layer-specific behavior when it comes to inductive ability?
* Is there a way to empirically verify the claims in Sec 6.2?
Appendix D: How does the feature mapping between RWKV and Mamba look like?
* Sec 6.1: Is the layer-17 phenomenon robust to random initializations? I.e., if one retrains the SSM with another seed, would layer 17 still be the key in induction?

---

> ### Author Response · Authors · 2024-11-23
> **Response to Reviewer nuP5 (1 / 6)**
>
> We sincerely appreciate your recognition of our work, with highlights in novelty, soundness and significance. We would also like to acknowledge the thorough and constructive feedback and questions provided that help strengthen our work, which we summarize as follows:
> - Clarifying the role of circuit analysis.
> - Empirically supporting the "Universal Induction Algorithm" claim.
> - Generalization to Larger Models
> - Universal layer-specificity phenomenon.
> - RWKV-Mamba feature similarity.
>
> All of these aspects are helpful and insightful. In addition, thanks for pointing out the typos in our submitted manuscript, which we have all fixed in our updated version.

---

> ### Author Response · Authors · 2024-11-23
> **Clarifying the Role of Circuit Analysis (2 / 6)**
>
> >The role of circuit analysis experiments is unclear.
>
> >They are indeed very interesting but I’m not sure how they contribute to building a mechanistic analogy between SSMs and Transformers.
>
> Thanks for this question. Our thoughts on this question are mostly influenced by a lot of existing discussion of features and circuits in the field of Mechanistic Interpretability[1, 2]. **We think that circuit analysis is investigating how features are connected from a more macroscopic viewpoint**. If we have found that interpretable features in different models are universal, a natural research question to ask next is whether the weights connect them in the same way.
>
> **We conjecture the main reason we left you confused is that our circuit analysis seems not correlated to the SAE features** investigated in Section 4 and 5. We are actually studying individual mamba blocks, SSM states or attention heads rather than what we claim to be "how the features are connected".
>
> If this is the case, we think that SAE feature circuits, despite some existing literature investigating this problem[1, 2, 3], is still not what we think a mature method to help support our claims due to its time and computation complexity. In addition, **we expect findings of SAE feature circuits to be a finer-grained extension of head-level or block-level circuit analysis**, which may not contradict with our findings.
>
> [1] [Dictionary Learning Improves Patch-Free Circuit Discovery in Mechanistic Interpretability: A Case Study on Othello-GPT](https://arxiv.org/abs/2402.12201)
>
> [2] [Sparse Feature Circuits: Discovering and Editing Interpretable Causal Graphs in Language Models](https://arxiv.org/abs/2403.19647v1)
>
> [3] [Transcoders Find Interpretable LLM Feature Circuits](https://arxiv.org/abs/2406.11944v1)

---

> ### Author Response · Authors · 2024-11-23
> **Empirically Supporting the "Universal Induction Algorithm" Claim. (3 / 6)**
>
> >The claims made in Section 6.2 need to be empirically supported.
>
> Thanks for this suggestion. We are not sure we currently understand your suggestion correctly. We take it as "designing extra experiments to show that induction circuits are similar".
>
> It is indeed necessary to quantify or validate cross-architectural circuit similarity with counterfactual experiments to further strengthen our claim that Mamba and Transformer implement the same induction algorithm. For instance, it may be a statistical method to reveal that model weights connecting similar features also tend to be similar. Or one can ablate matched pairs of features from both models and see whether downstream performance drops in the same trend. However, due to time and compute constraint, we are not currently able to conduct such experiments. Apologies for this. Nonetheless, since induction circuits have been widely studied for Transformers[1, 2], **we think that the "previous token heads-local convolution" and "Induction heads-Layer 17 SSM State" analogies are strongly backed up this claim**.
>
> [1][A Mathematical Framework for Transformer Circuits](https://transformer-circuits.pub/2021/framework/index.html)
>
> [2][In-context Learning and Induction Heads](https://transformer-circuits.pub/2022/in-context-learning-and-induction-heads/index.html)

---

> ### Author Response · Authors · 2024-11-23
> **Universal Layer-Specificity Phenomenon (4 / 6)**
>
> >Do Transformers have such layer-specific behavior when it comes to inductive ability?
>
> >Is the layer-17 phenomenon robust to random initializations? I.e., if one retrains the SSM with another seed, would layer 17 still be the key in induction?
>
> Thanks for this interesting question. It is an important hypothesis that the inter-layer structure of the final model should be very robust to initialization and even more other hyperparameters. **We conjecture layer 17 will still be a pivotal induction layer with a retrained Mamba**. One of the main reasons is that two Transformers trained with mostly the same configuration by two groups, [Pythia-160M](https://huggingface.co/EleutherAI/pythia-160m/blob/main/config.json) and [GPT2-Small](https://huggingface.co/openai-community/gpt2/blob/main/config.json), **have both been reported to mainly use their layer 5 (out of 12 layers) to perform induction**.
> - Both models have a hidden dimension D=768 and # layers = 12. The main differences are summarized as follows:
> | Model          | Position Embedding    | Embedding & Unembedding | Trained with Dropout |
> |-----------------|-----------------------|--------------------------|-----------------------|
> | GPT2-Small      | Absolute, sinusoidal | Tied                     | Yes                   |
> | Pythia-160M     | Rotary               | Independent              | No                    |
> - GPT2-Small: We quote Section 4.2 in [1], "As a case study, we focus on GPT-2 Small [55], which has two induction heads in layer 5 (heads 5.1 and 5.5) "
> - Pythia-160M: We perform path patching on this model, finding that head 5.0 (the first attention head in layer 5) is the most notable induction heads:
> | layer\head | 0     | 1     | 2     | 3     | 4     | 5     | 6     | 7     | 8     | 9     | 10    | 11    |
> |--------|-------|-------|-------|-------|-------|-------|-------|-------|-------|-------|-------|-------|
> | 0          | 0.00  | 0.00  | 0.00  | 0.00  | 0.00  | 0.00  | 0.00  | 0.00  | 0.00  | 0.00  | 0.00  | 0.00  |
> | 1          | 0.07  | -0.15 | -0.10 | 0.03  | 0.09  | -0.08 | -0.07 | 0.06  | -0.01 | 0.11  | 0.34  | -0.05 |
> | 2          | -0.14 | 0.07  | 0.10  | 0.14  | 0.14  | -0.13 | 0.60  | -0.03 | -0.14 | 0.10  | 0.04  | 0.03  |
> | 3          | -0.24 | -0.14 | -0.96 | -1.20 | -0.49 | -0.14 | 0.20  | -0.38 | -0.10 | 0.06  | -0.11 | -0.07 |
> | 4          | 0.13  | -0.26 | 0.09  | -0.16 | -0.10 | -0.02 | 0.89  | 0.13  | 0.09  | -0.28   | -0.14   | 0.30  |
> | 5          | **4.00**  | -0.20 | 0.05  | 0.06  | -0.53 | -0.04 | 0.48  | 0.62  | 0.06  | 0.08  | 0.05  | -0.23 |
> | 6          | -0.04 | -0.23 | -0.04 | -0.22 | 0.02  | 0.09  | 0.04  | -0.33 | 0.02  | -0.04 | -0.38 | 0.04  |
> | 7          | -0.28 | 0.17  | 0.03  | 0.06  | -0.28 | -0.07 | 0.01  | -0.18 | -0.23 | -0.03 | -0.02 | 0.18  |
> | 8          | -0.07 | 0.03  | 0.50  | 0.00  | 0.15  | -0.02 | 0.01  | -0.22 | 0.02  | -0.02 | -0.08 | 0.38  |
> | 9          | 0.54  | -0.03 | 0.07  | -0.09 | -1.10 | -0.04 | 0.04  | 0.00  | 0.04  | 0.10  | -0.01 | 0.02  |
> | 10         | -0.01 | 0.03  | 0.00  | 0.00  | -0.03 | -0.10 | 0.01  | -0.01 | 0.00  | -0.04 | 0.03  | 0.01  |
> | 11         | -0.14 | -0.13 | -0.05 | -0.04 | 0.00  | -0.02 | -0.11 | -0.02 | 0.01  | -0.07 | -0.02 | 0.06  |
> [1] [Interpreting Attention Layer Outputs with Sparse Autoencoders](https://arxiv.org/pdf/2406.17759v1)

---

> ### Author Response · Authors · 2024-11-23
> **RWKV-Mamba Feature Similarity (5 / 6)**
>
> >How does the feature mapping between RWKV and Mamba look like?
>
> Thanks for this question. we provide **Pythia-160m&Mamba-130m&RWKV-169m similarity results** as shown below(mean MPPC of A->B):
>
> | Model A / Model B | Pythia | Mamba | RWKV |
> |------------------|--------|-------|------|
> | **Pythia**       | 1      | 0.68  | 0.61 |
> | **Mamba**        | 0.74   | 1     | 0.71 |
> | **RWKV**         | 0.49   | 0.55  | 1    |
>
>
> **We have incorporated these updates in Sections 4.4 and Appendix D.5 of the revised manuscript** for further clarity and elaboration.

---

> ### Author Response · Authors · 2024-11-23
> **Generalization to Larger Models (6 / 6)**
>
> >(Reviewer 4GEc) How do you expect your findings to scale to larger models? Did you observe whether model size impacts universality between architectures? Could smaller or larger versions of Transformers and Mambas exhibit different degrees of feature similarity?
>
> >(Reviewer nuP5) (Minor) The size of LMs is limited. Only ~100m models are used in the experiments.
>
> >(Reviewer jsst) Without further experiments, it cannot be excluded that the limited capacity of tiny models might be the main motivation behind the high similarity features and circuits across the two architectures, and this could not be the case for more capable models with e.g. 1B or 7B parameters.
>
> Thanks for pointing this out, which greatly helps improve our work.
>
> We additionally conduct the experiment for **2.8B variants of both models**, giving us the following results:
> | Mean MPPC / Model Size       | 130M (original) | 2.8B  |
> |------------------------------|-----------------|-------|
> | Main experiment              | 0.681           | 0.792 |
> | Skyline 1 (Model Seed Variant) | *0.725*           | *0.847* |
> | Skyline 2 (SAE Seed Variant)   | **0.806**           | **0.878** |
>
> **We have incorporated these updates in Sections 4.4 and Appendix D.4** of the revised manuscript for further clarity and elaboration.

---

> ### Author Response · Authors · 2024-12-01
> **Follow-up on Feedback and Rating for Submission 4540**
>
> Dear Reviewer nuP5,
>
> We hope this email finds you well. We are reaching out regarding our ICLR 2025 submission (Submission 4540). With the discussion period coming to a close, We wanted to follow up and see if there are any additional questions or concerns about our paper or the rebuttal we provided earlier that we could help clarify.
>
> Your detailed feedback has been Insightful, and we have put significant effort into addressing the points you raised. If there are any remaining aspects where further clarification might strengthen your understanding of our work, please let us know—we would be happy to provide more information.
>
> Furthermore, if you feel our responses have addressed your concerns effectively, we would greatly appreciate it if you might consider revisiting your initial rating of our submission. Your expert evaluation plays a crucial role in shaping the final outcome, and we sincerely appreciate your time and efforts throughout this review process.
>
> Thank you again for your dedication to improving the quality of submissions. Please feel free to let us know if there is anything else we can assist with.
>
> Best regards,
>
> Authors of Submission 4540

---

### Official Review · Reviewer_4GEc · 2024-11-07

**Soundness:** 2
**Presentation:** 2
**Contribution:** 2
**Rating:** 6
**Confidence:** 3

**Summary:**

The paper investigates the "universality hypothesis" in mechanistic interpretability, which suggests that different neural network architectures may converge to implement similar algorithms when tasked with analogous objectives. The authors focus on two mainstream architectures for language modeling: Transformers and Mambas. They propose using Sparse Autoencoders (SAEs) to extract interpretable features from these models and demonstrate that a significant portion of features are shared between the two architectures. The paper validates the correlation between feature similarity and universality and delves into the circuit-level analysis of Mamba models, finding structural analogies with Transformers, particularly in induction circuits.

The paper's contributions include:
- Introduction of a novel metric to isolate and quantify feature universality in the context of architectural variations.
- Empirical evidence shows that Transformer and Mamba models learn similar features through the application of SAEs.
- Circuit analysis of Mamba models reveals structural analogies and nuanced differences compared to Transformer circuits.
- Support for the universality hypothesis by demonstrating cross-architecture feature similarity and identifying the "Off-by-One motif" in Mamba models.

**Strengths:**

- The identification of the "Off-by-One motif" in Mamba models is a unique contribution that highlights nuanced differences between architectures.

- The introduction of a complexity-based interpretation for understanding feature similarity differences is innovative.

- The circuit-level analysis of Mamba models, revealing structural analogies with Transformers, is good and adds depth to the study. The validation of feature similarity and its correlation with universality further strengthens the study.


- The findings of this paper have implications for the field of neural network interpretability. By demonstrating that different architectures can converge to similar algorithms and features, the study provides valuable insights into the generalizability of mechanistic findings across models.

**Weaknesses:**

- While the paper focuses on Transformers and Mambas, it would benefit from a broader examination of additional architectures. Including a more diverse set of models, such as recurrent neural networks (RNNs) or convolutional neural networks (CNNs), could strengthen the universality hypothesis by offering a more comprehensive understanding of feature similarity across a wider range of neural networks. This would enhance the generalizability of the findings.

- The paper utilizes OpenWebText for correlation analysis but does not discuss how the choice of dataset might affect the results. A more detailed examination of the potential biases and limitations introduced by the dataset choice would provide a clearer context for the findings and ensure that the results are not overly dependent on a specific dataset.

- The claims of feature similarity and universality would be more robust if supported by statistical significance tests. Including such tests would provide stronger evidence for the observed correlations and enhance the credibility of the conclusions.

- SAE-related technical gaps:
    - The paper does not include ablation studies on SAE hyperparameters (dictionary/code size, training duration, etc.). Conducting these studies would help to understand the sensitivity of the results to different hyperparameter settings and ensure the robustness of the findings.
    - There is no discussion of how SAE reconstruction quality relates to feature similarity. Addressing this relationship would provide insights into the effectiveness of SAEs in isolating interpretable features and validate the methodology used.

- The use of GPT-4 for complexity scoring lacks rigorous validation. The paper does not provide inter-rater reliability metrics or comparisons with human annotations, nor does it discuss potential biases in the automated scoring.

- The paper provides a limited exploration of why the "Off-by-One" motif exists in Mamba models. A deeper investigation into the underlying reasons for this motif would enhance the understanding of the structural differences between the Mamba and Transformer models and provide more insights into the universality hypothesis.

**Questions:**

1. Why did you choose OpenWebText as your primary dataset for analysis? How might the choice of OpenWebText as the dataset influence your results? Have you tested if the feature similarities hold across different domains (e.g., code, mathematics, or structured data)? Would analyzing domain-specific text reveal different patterns of architectural universality?

2. Have you performed any statistical significance tests to support your claims of feature similarity and universality?

3. How generalizable are your findings to other tasks beyond language modeling?

4. Can you provide more details on why the "Off-by-One" motif exists in Mamba models?

5. Is there a risk that the Sparse Autoencoder pre-processing itself may impose a degree of alignment between features in Transformers and Mambas? Could the sparsity constraint inadvertently enhance apparent similarity?

6. How do you expect your findings to scale to larger models? Did you observe whether model size impacts universality between architectures? Could smaller or larger versions of Transformers and Mambas exhibit different degrees of feature similarity?

---

> ### Author Response · Authors · 2024-11-23
> **Response to Reviewer 4GEc (1 / 9)**
>
> We sincerely appreciate your recognition of our work, with highlights in novelty, quality, depth, and significance. We would also like to acknowledge the thorough and constructive feedback and questions provided which help strengthen our work, which we summarize as follows:
> - Statistical significance tests;
> - Generalization to larger models and other tasks;
> - Ablation studies on SAE hyperparameters;
> - Manually validating Autointerp complexity and monosemanticity scores;
> - Dataset Choice;
> - Further Exploring Off-by-one motif;
> - A broader examination of additional architectures;
> - Could the sparsity constraint inadvertently enhance apparent similarity?
>
> All of these aspects are helpful and insightful. We sorted them in the order we think are of decreasing importance and respond in the following comments:

---

> ### Author Response · Authors · 2024-11-23
> **Statistical Significance Tests (2 / 9)**
>
> >(Reviewer 4GEc) Have you performed any statistical significance tests to support your claims of feature similarity and universality?
>
> >(Reviewer jsst) In Section 4.4, no hypothesis is formulated regarding the expected similarity of features found across the four tested variants, and consequently, no significance measure for the correlation coefficients is reported.
>
> This is an important part we missed in our submitted manuscript and we are very thankful for pointing this out.
>
> We additionally establish a **random baseline** by calculating MPPC for each Pythia feature against a random SAE **with matching feature count and sparsity level** (by masking all but the TopK activating features) to the Mamba SAE to observe the impact of feature quantity and sparsity on MPPC distribution.
>
> We denote mean MPPC as the random variable \\( x \\). To evaluate whether the result of our main experiment (\\( x = 0.681 \\)) belongs to a distribution with the same mean as the random baseline, we conducted a **Hypothesis Testing**. Specifically, we tested the null hypothesis (\\( H_0 \\)) that **the mean of the experimental group is equal to the mean of the random baseline group** (\\( \mu_{\text{experiment}} = \mu_{\text{baseline}} \\)) against the alternative hypothesis (\\( H_1 \\)) that the two means are different (\\( \mu_{\text{experiment}} \neq \mu_{\text{baseline}} \\)).
>
> The random baseline group consisted of 16 samples (repeated the baseline 16 times and calculate mean MPPC for each) with a mean of \\( 0.1944 \\) and a standard deviation of \\( 0.00046 \\). We made the simplifying assumption that the variance of the experimental group is the same as that of the baseline group. The experimental group consisted of a single sample with \\( x = 0.681 \\). Since the sample size of the experimental group is one, we applied a two-sample \\( t \\)-test using the pooled standard deviation from the baseline group. Yielding a \\( t \\)-value of \\( -1026.24 \\). The degrees of freedom are \\( n_{\text{baseline}} + n_{\text{experiment}} - 2 = 15 \\). Using a **one-tailed \\( t \\)-test**, the resulting \\( p \\)-value is:
>
> $p = 4.54 \times 10^{-38}$
>
> Given that the \\( p \\)-value is significantly smaller than any conventional significance level (\\( \alpha = 0.05 \\)), we **reject the null hypothesis**. This indicates that the experimental result (\\( x = 0.681 \\)) is highly unlikely to belong to a distribution with the same mean as the random baseline.
>
> **We have incorporated these updates in Sections 4.1 and 4.4 of the revised manuscript** for further clarity and elaboration.

---

> ### Author Response · Authors · 2024-11-23
> **Generalization to Larger Models and Other Tasks (3 / 9)**
>
> >(Reviewer 4GEc) How do you expect your findings to scale to larger models? Did you observe whether model size impacts universality between architectures? Could smaller or larger versions of Transformers and Mambas exhibit different degrees of feature similarity?
>
> >(Reviewer nuP5) (Minor) The size of LMs is limited. Only ~100m models are used in the experiments.
>
> >(Reviewer jsst) Without further experiments, it cannot be excluded that the limited capacity of tiny models might be the main motivation behind the high similarity features and circuits across the two architectures, and this could not be the case for more capable models with e.g. 1B or 7B parameters.
>
> Thanks for pointing this out, which greatly helps improve our work.
>
> We additionally conduct the experiment for **2.8B variants of both models**, giving us the following results:
> | Mean MPPC / Model Size       | 130M (original) | 2.8B  |
> |------------------------------|-----------------|-------|
> | Main experiment              | 0.681           | 0.792 |
> | Skyline 1 (Model Training Variant) | *0.725*           | *0.847* |
> | Skyline 2 (SAE Seed Variant)   | **0.806**           | **0.878** |
>
> **We have incorporated these updates in Sections 4.4 and Appendix D.4** of the revised manuscript for further clarity and elaboration.
>
> >How generalizable are your findings to other tasks beyond language modeling?
>
> We do not further investigate this problem since it is slightly beyond the scope of this work. Nonetheless, we are optimistic about the generalizability for the following reasons.
> - Our method (i.e., Sparse Autoencoders) has been shown to generalize to vision models[1, 2], Othello & chess models[3, 4] and protein language models[5] etc.
> - There has been lines of evidences that interpretable vision neurons[6] and circuits[7] can be observed across a variety of vision model architectures. However, there is possibility that simpler features are neuron-aligned and more complex ones are stored in superposition and turn out do not match across architectures. We are also excited to see this line of work continue to ViT and conv model universality and we currently expect them to be similar as well.
>
> [1] [https://livgorton.com/inceptionv1-mixed5b-sparse-autoencoders/](https://livgorton.com/inceptionv1-mixed5b-sparse-autoencoders/)
>
> [2] [Towards Multimodal Interpretability: Learning Sparse Interpretable Features in Vision Transformers](https://www.lesswrong.com/posts/bCtbuWraqYTDtuARg/towards-multimodal-interpretability-learning-sparse-2)
>
> [3] [Dictionary Learning Improves Patch-Free Circuit Discovery in Mechanistic Interpretability: A Case Study on Othello-GPT](https://arxiv.org/abs/2402.12201)
>
> [4] [Evaluating Sparse Autoencoders with Board Game Models](https://www.lesswrong.com/posts/EWhA4pyfrbdSkCd4G/evaluating-sparse-autoencoders-with-board-game-models)
>
> [5] [InterPLM: Discovering Interpretable Features in Protein Language Models via Sparse Autoencoders](https://www.biorxiv.org/content/10.1101/2024.11.14.623630v1.full.pdf)
>
> [6] [Zoom in: An Introduction to Circuits](https://distill.pub/2020/circuits/zoom-in/#claim-3)
>
> [7] [High-Low Frequency Detectors](https://distill.pub/2020/circuits/frequency-edges/#universality)

---

> ### Author Response · Authors · 2024-11-23
> **Ablating SAE Training Hyperparameters / Settings (4 / 9)**
>
> >The paper does not include ablation studies on SAE hyperparameters (dictionary/code size, training duration, etc.). There is no discussion of how SAE reconstruction quality relates to feature similarity.
>
> We appreciate your constructive feedback. We have included more ablations studies on dictionary size, L1 coefficient and SAE architecture (TopK variant). The results are as follows:
>
> **L1 coefficient**:
> | Mean MPPC / L1                  | 1e-4             | 2e-4 (Original)   | 4e-4             |
> |---------------------------------|------------------|-------------------|------------------|
> | Main experiment                 | 0.673          | 0.681           | 0.720          |
> | Skyline 1 (Model Seed Variant)  | *0.717*        | *0.725*        | *0.766*        |
> | Skyline 2 (SAE Seed Variant)    | **0.800**            | **0.806**             | **0.833**            |
>
> **Dictionary size (Expansion factor * model hidden size D)**:
> | Mean MPPC / Dictionary Size F   | 32 * 768 (Original) | 64 * 768 |
> |---------------------------------|---------------------|----------|
> | Main experiment                 | 0.681               | 0.701    |
> | Skyline 1 (Model Seed Variant)  | *0.725*               | *0.733*    |
> | Skyline 2 (SAE Seed Variant)    | **0.806**               | **0.795**    |
>
> We do not include results with respect to training duration because our SAEs quickly converge after 30 minutes of training on an H100 GPU. Due to time and computation constraint we do not further ablate on this setting.
>
> **We have incorporated these updates in Sections 4.4 and Appendix D.1 of the revised manuscript** for further clarity and elaboration.

---

> ### Author Response · Authors · 2024-11-23
> **Manually Validating Autointerp Scores (5 / 9)**
>
> >The use of GPT-4 for complexity scoring lacks rigorous validation. The paper does not provide inter-rater reliability metrics or comparisons with human annotations, nor does it discuss potential biases in the automated scoring.
>
> Thanks for pointing this out. Without rigorously validating the reliability of autointerp scores, it is questionable to draw the conclusion in Section 5. We ask **two human annotators to score** 64 (out of 358 automatically evaluated) pairs for complexity scores and 32 (out of 72) for monosemantic scores. We **take the average of human labeled scores** and fit human-GPT-4 consistency. The fitted results in terms of (scope, R-square score) is (scope=0.45, R2=0.29) for complexity and (scope=0.38, R2=0.17) for monosemanticity, suggesting the existence of human-GPT4 scoring consistency. We notice that compared to GPT-4 labeled scores, human annotators tend to be more polarized, which may cause a lower R-square score.
>
> **We have incorporated these updates in Sections 5.2 and Appendix I of the revised manuscript** for further clarity and elaboration.

---

> ### Author Response · Authors · 2024-11-23
> **Dataset Choice (6 / 9)**
>
> >Why did you choose OpenWebText as your primary dataset for analysis? How might the choice of OpenWebText as the dataset influence your results? Have you tested if the feature similarities hold across different domains (e.g., code, mathematics, or structured data)? Would analyzing domain-specific text reveal different patterns of architectural universality?
>
> Thanks for these insightful questions. The main reason we choose OWT as our primary dataset is that **OWT is a widely used comprehensive text corpus** in this field[1, 2, 3]. This is probably because of the popularity of GPT2-Small, whose training data WebText has its open-sourced version OWT.
>
> It can be the case that older datasets of lower quality and duplication can result in overhigh correlation. To address this, **we additionally perform correlation analysis on two more datasets**, namely SlimPajama[4], a more recent cleaned and deduplicated text corpus for pretraining, and Github subset of RedPajama[5] for domain-specific analysis. The results are shown as follows:
>
> | Mean MPPC / Dataset             | OpenWebText (Original) | SlimPajama | RedPajama-Github subset |
> |---------------------------------|------------------------|------------|-------------------------|
> | Main experiment                 | 0.74                  | 0.681      | 0.745                   |
> | Skyline 1 (Model Seed Variant)  | *0.76*                  | *0.725*      | *0.782*                   |
> | Skyline 2 (SAE Seed Variant)    | **0.81**                 | **0.806**      | **0.806**                   |
>
>
> We again appreciate this question as it makes us decide to demonstrate all of our experimental results with ones obtained on SlimPajama, rather than OWT, for its comprehensiveness, higher quality and containing OOD text data for both models we tested. This helps improve the robustness of our conclusion.
>
> **We have incorporated these updates in Sections 4.4 and Appendix D.2 of the revised manuscript** for further clarity and elaboration.
>
> [1] [Interpretability in the Wild: a Circuit for Indirect Object Identification in GPT-2 small](https://arxiv.org/abs/2211.00593)
>
> [2] [Transcoders Find Interpretable LLM Feature Circuits](https://arxiv.org/abs/2406.11944v1)
>
> [3] https://www.alignmentforum.org/posts/f9EgfLSurAiqRJySD/open-source-sparse-autoencoders-for-all-residual-stream
>
> [4] https://cerebras.ai/blog/slimpajama-a-627b-token-cleaned-and-deduplicated-version-of-redpajama
>
> [5] https://www.together.ai/blog/redpajama

---

> ### Author Response · Authors · 2024-11-23
> **Further Exploring Off-by-one motif (7 / 9)**
>
> >Can you provide more details on why the "Off-by-One" motif exists in Mamba models?
>
> This is an important question to understand circuits in Mamba models. However, our conjecture about this phenomenon is not rigorously tested and we leave this for future work focusing on Mamba circuits. Our current hypothesis on this problem is that Mamba performs "Off-by-One" to **implicitly revert the write-read order of its SSM States**.
>
> **Write-and-Read Nature of SSMs**:
> Write: $h_{i}^{(l)} = F_a(c_{i}^{(l)}) \circ h_{i-1}^{(l)} + F_b(c_{i}^{(l)})$.
> Read: $s_i^{(l)} = h_i^{(l)} (W_c^{(l)} * c_i^{(l)}) + W_d^{(l)} \circ c_i^{(l)}$.
> Concretely, an SSM block has three data-dependent transformations A, B, C and a shortcut D. **It first write its input at token $i$** $c_i^{(l)}$ to the state space multiplied by B ($F_b(c_{i}^{(l)})$) and add this to its past state with a transformation $F_a(c_{i}^{(l)}) \circ h_{i-1}^{(l)}$. **It then reads** from this state space with matrix C plus a shortcut, where the input $c_i^{(l)}$ is directly transformed by D without interacting with the SSM state.
>
> **The gate branch $g$ might prevent the need to write into SSM states in-time**.
> There are two ways for a Mamba block input, at step $i$ at layer $l$, $x_i^l$  to contribute to the block output: via the gate branch $g_i^l$ and via the conv-SSM branch. If the local convolution only merges information from the past timesteps rather than the current time step $i$,  this means **the conv-SSM branch are blocked for input $x_i^l$**. However, $x_i^l$ can still affect subsequent computation via the gate. And it is not written into the SSM state until the next timestep $i+1$ due to Off-by-One.
>
> **This implicitly reverts the write-read order of its SSM States for easier token-mixing**.
> In this case where the conv-SSM branch are blocked for the current timestep $i$, the SSM actually **first reads from information in the past, and write information in $x_i^l$ in the next step**. We conjecture this is beneficial in reducing one term in the state space so that it can more effectively retrieve past information.
>
> Since there is relatively less work on Mamba interpretability compared to Transformers, we are open to the possibility that we missed something or made improper assumptions here. We do not dive deeper into this problem due to time and compute limitation and its being slightly beyond the scope of this paper.

---

> ### Author Response · Authors · 2024-11-23
> **A Broader Examination of Additional Architectures (8 / 9)**
>
> >Including a more diverse set of models, such as recurrent neural networks (RNNs) or convolutional neural networks (CNNs), could strengthen the universality hypothesis by offering a more comprehensive understanding of feature similarity across a wider range of neural networks. This would enhance the generalizability of the findings.
>
> Thanks for the reasonable and consturctive suggestion. We would like to point out that **we included results for Pythia-RWKV** in Appendix D of our submitted manuscript, which is known as an RNN-like language model architecture. We have made this clearer in our revised version. In addition, we provide **Pythia-160m&Mamba-130m&RWKV-169m similarity results** as shown below(mean MPPC of A->B):
>
> | Model A / Model B | Pythia | Mamba | RWKV |
> |------------------|--------|-------|------|
> | **Pythia**       | 1      | 0.68  | 0.61 |
> | **Mamba**        | 0.74   | 1     | 0.71 |
> | **RWKV**         | 0.49   | 0.55  | 1    |
>
> It is also reasonable to include more novel architectures like xLSTM, RetNet or even convolution-based language model architectures and will indeed enhance the generalizability of our findings. Currently, however, we do not have resources at hand to conduct these experiments and we will leave this for future work. Apologies for this.
>
> **We have incorporated these updates in Sections 4.4 and Appendix D.5 of the revised manuscript** for further clarity and elaboration.

---

> ### Author Response · Authors · 2024-11-23
> **Could the Sparsity Constraint Inadvertently Enhance Apparent Similarity? (9 / 9)**
>
> >Is there a risk that the Sparse Autoencoder pre-processing itself may impose a degree of alignment between features in Transformers and Mambas? Could the sparsity constraint inadvertently enhance apparent similarity?
>
> We appreciate your insightful question. This is indeed a reasonable and inspiring concern.
>
> There are two possibilities we can think of that this interpretability illusion holds: (1) **Illusion in MPPC analysis**: sparsity constraint causes the activation pattern of all features to be sparse, i.e., activating only on 1% of all tokens, leading to an increase in Max Pairwise Pearson Correlation. This turns out to be the case in the random baseline we established in Response 2/9, compared to one without being set to the same the sparsity level as a normal SAE. Nonetheless, it accounts for only a small portion of Pythia-Mamba feature similarity, as indicated by the statistical significance test. (2) **Illusion in SAE Training**, which we discuss as follows:
>
> We think there is possibility that different model architectures actually learn different sets of features but SAEs inadvertently aligns them. **If SAEs learn commonly-composed features, it may be the case that we are being over-confident about feature universality**. It has already been suggested by [1] that SAEs may learn compositions of the "true" underlying features in a toy model due to the sparsity constraint. For instance, if two models encode the same concept in different ways (e.g. one identifies dogs with color and another identifies with breeds) but our SAEs are too small to capture such difference and only learn a "universal dog feature", we are then fooled and draw the wrong conclusion.
>
> There are two reasons why we are not quite concerned with this possibility. (1) **Holding all else equal, larger SAEs exhibit higher MPPC values (Response 4/9)**. This serves as negative evidence against the hypothesis above since if features of larger SAEs 'split' into different sub-features, there should be a drop in MPPC. We are also open to the possibility that we have not scale our SAEs enough yet.  (2) Even if there exists interpretability illusion for some reason we currently are not aware of, our findings reliably suggest **there at least exists a universal, sparse and interpretable, though lossy, decomposition of the models' hidden activations**. It is an exciting topic to discover hard-to-notice divergence hiding in part of the activation space uncaptured by SAEs, probably with better SAE training techniques[2] or some well-designed probes.
>
> **We have incorporated the baseline updates in Sections 4.1 and 4.4 of the revised manuscript** for further clarity and elaboration.
>
> [1] https://www.lesswrong.com/posts/a5wwqza2cY3W7L9cj/sparse-autoencoders-find-composed-features-in-small-toy
>
> [2] [Sparse Crosscoders for Cross-Layer Features and Model Diffing](https://transformer-circuits.pub/2024/crosscoders/index.html)

---

> ### Author Response · Authors · 2024-12-01
> **Follow-up on Feedback and Rating for Submission 4540**
>
> Dear Reviewer 4GEc,
>
> We hope this email finds you well. We are writing to follow up regarding our submission (Submission Number: 4540). We noticed that the discussion period is nearing its end, and We wanted to kindly check if you had any further questions or concerns regarding our paper or the rebuttal we provided earlier.
>
> We deeply value your feedback and have aimed to address your comments comprehensively in our rebuttal. If there are any remaining points that we could clarify or elaborate on, please do not hesitate to let us know.
>
> Additionally, if you believe our responses sufficiently addressed your concerns, we kindly ask if you would consider revisiting your rating for our submission. Your assessment is incredibly important to us, and we appreciate the time and effort you have dedicated to reviewing our work.
>
> Thank you once again for your thoughtful review and contributions to improving our paper. Please feel free to reach out with any further questions or comments.
>
> Best regards,
>
> Authors of Submission 4540

---

> > ### Comment · Reviewer_4GEc · 2024-12-02
> >
> > Thanks for the new experiments and replies to my concerns. I have updated my score. Good luck!

---

> > > ### Author Response · Authors · 2024-12-03
> > > **Response to Reviewer 4EGc: Thank You for Your Feedback**
> > >
> > > Thank you for taking the time to review our responses and the additional experiments. We greatly appreciate your feedback and support throughout the review process. Best wishes!

---

### Author Response · Authors · 2024-12-04
**General Response**

We sincerely thank all reviewers for their thoughtful feedback and constructive suggestions, which have greatly helped us refine this work. Below, we summarize and address the key points raised:

- We are grateful for the recognition from all four reviewers for the **significance** (reviewers 4GEc, nuP5, eqTo, and jsst), **soundness** (reviewers 4GEc, nuP5, eqTo, and jsst), and **novelty** (reviewers 4GEc, nuP5, and eqTo) of our work.
- Specifically, we appreciate the recognition of our **complexity-based perspective on feature similarity** as both innovative (reviewer 4GEc) and interesting (reviewer jsst).
- The **skylines introduced for comparison** were also highlighted as well-motivated and supportive (reviewers jsst and eqTo).
- Additionally, our **induction circuit universality analysis** was acknowledged as adding depth to the study (reviewer 4GEc).

We are particularly thankful for the reviewers’ insightful suggestions, which have significantly enhanced this work. Below, we address four common concerns:

1. **Generalization to larger models** (reviewers 4GEc, nuP5, and jsst):
   Our initial experiments were conducted on ~100M parameter models, and we acknowledge concerns about scaling to modern, larger models. To address this, we conducted our main experiments along with corresponding baselines and skylines on 2.8B versions of both models. These results align with the trends reported in the original manuscript, demonstrating the robustness of our findings at larger scales.

2. **Statistical significance for robust comparison** (reviewers 4GEc and jsst):
   To ensure the reliability of our results, we conducted statistical significance tests, confirming a probability of less than $5 \times 10^{-38}$ for random noise influence. Additionally, cross-architecture comparisons on complex features revealed greater similarities than cross-layer comparisons, highlighting the robustness and meaningfulness of our analysis.

3. **Ablation studies on SAE training settings** (reviewers 4GEc and jsst):
   Reviewers expressed concerns about the robustness of our findings to variations in SAE training hyperparameters and architectures. We expanded our ablation studies to include sparsity coefficients, SAE size, and TopK SAEs. These additional experiments confirm the robustness of our conclusions across a broader range of training settings.

4. **Further discussions** (reviewers 4GEc, eqTo, and jsst):
   During the discussion period, several inspiring ideas emerged that we plan to incorporate into future versions of this work. These include:
   - **Implications of complexity-based MPPC analysis** for broader model comparison methods (from discussions with reviewer jsst).
   - **Further details on the Mamba Off-by-One motif** (suggested by reviewer 4GEc).
   - **Highlighting an interesting exception** in the final-layer representation observed in our depth-specialization analysis (suggested by reviewer eqTo).

These insights add significant depth to our work, and we thank the reviewers for their valuable contributions. We also deeply value additional feedback not explicitly mentioned here, which we address in the individual discussion sections. We are committed to incorporating these suggestions to improve both the depth and clarity of our work.

---

### Meta-Review · Area_Chair_74W2 · 2024-12-19

**Metareview:**

The paper investigates the universality hypothesis, showing that Transformers and Mambas, despite architectural differences, learn similar features for language modeling tasks. Using Sparse Autoencoders (SAEs), the authors demonstrate cross-architecture feature similarity and identify a novel “Off-by-One motif” in Mamba models, providing new insights into their induction circuits. The study combines feature-level analysis and circuit-level comparisons to support claims of mechanistic universality.

The strengths of the paper lie in its novel application of SAEs to compare architectures, robust empirical results validated by statistical significance tests, and the depth of analysis that identifies meaningful similarities and differences. The primary weaknesses include limited experiments on larger models and questions about the robustness of the results. Additionally, reviewers raised concerns about potential biases introduced by SAE pre-processing and the generalization of findings beyond language modeling. Most of these issues were addressed during the rebuttal period: the authors provided results for larger models (2.8B parameters), performed statistical tests, and demonstrated that the SAE method did not artificially impose alignment. While broader generalization and deeper circuit validation remain future directions, the authors' responses sufficiently strengthen their claims.

Overall the paper makes a significant and timely contribution to neural network interpretability. Its findings are well-supported, and the concerns raised have been addressed to a reasonable extent. I recommend accepting this paper.

**Additional Comments On Reviewer Discussion:**

During the rebuttal period, several important points were raised by the reviewers, and the authors addressed these through additional experiments and clarifications.

Reviewer 4GEc raised concerns about the scalability of the study, noting that the experiments were conducted on small models (~100M parameters). In response, the authors conducted additional experiments on larger models (2.8B parameters), which confirmed the robustness of their findings. This addressed the concern effectively, and the reviewer updated their score accordingly.

Reviewer nuP5 questioned the role of the circuit-level analysis and its contribution to building a mechanistic analogy between Transformers and Mambas, particularly regarding layer-specific behavior and inductive ability. They also asked whether the claims in Section 6.2 could be empirically supported. The authors acknowledged that while the circuit analysis was an evolving area, further experiments to validate the claims were beyond the scope of the work. They suggested these aspects could be explored in future research. This issue was noted, but it did not detract significantly from the paper's merit, given the other valuable contributions.

Reviewer 4GEc also raised the concern about the lack of statistical significance testing to support claims of feature similarity and universality. The authors responded by performing a two-sample t-test, showing a very low p-value and confirming that the observed similarities were statistically significant. This effectively addressed the concern and bolstered the credibility of their findings.

Reviewer nuP5 raised the possibility that the SAE pre-processing might have induced alignment between features in different models, potentially creating an "interpretability illusion." The authors responded by demonstrating that the similarity remained even when the sparsity constraint was adjusted. They argued that the findings were not driven by SAE-specific effects, which helped mitigate concerns about methodological biases.

Finally, Reviewer 4GEc asked about the generalization of the findings beyond language modeling to other tasks. While the authors did not extend their analysis to other domains, they referenced other work showing that their method could generalize well to tasks like vision and protein modeling.

In weighing these points for the final decision, I found the authors’ responses to be thorough and satisfactory. The additional experiments, particularly regarding model scalability and statistical significance, strengthened the paper significantly. While some concerns about the circuit analysis and SAE effects remain, these were acknowledged and set for future exploration, without undermining the overall contribution of the paper.

---

### Decision · Program_Chairs · 2025-01-22

Accept (Poster)